# 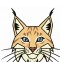 V-LynX: Token Interface Alignment for Video+X LLMs

**Jungin Park** [1]    **Jiyoung Lee** [2] [*]    **Kwanghoon Sohn** [1] [*]

## Abstract

This study introduces an intriguing phenomenon in Video LLMs: rather than merely translating frames into textual embeddings, Video LLMs establish a continuous manifold, *token interface*, allowing visual tokens to operate as standalone entities within the architecture. Exploiting this discovery, we propose V-LynX, a scalable framework that integrates novel modalities into Video LLMs by repurposing the internalized interface. Departing from conventional paradigms that necessitate heavy modality-specific encoders or paired supervision, V-LynX employs a lightweight auxiliary pathway in parallel with the frozen vision encoder. Our method integrates new sensory inputs with intrinsic video priors by aligning both attention responses and statistical distributions using unpaired unimodal data sets. This ensures manifold compatibility while preserving the integrity of the Video LLMs. Extensive benchmarks demonstrate that V-LynX achieves SOTA and efficiency across audio-visual QA, 3D reasoning, high-frame-rate, and multi-view video understanding. The code is available at project site.

## 1. Introduction

The advent of video large language models (Video LLMs) (Li et al., 2025a; Zhang et al., 2023; Cheng et al., 2024; Wang et al., 2024) highlights remarkable capabilities on sophisticated scene understanding by capturing long-range temporal dependencies. Nonetheless, despite their apparent multimodality, most existing Video LLMs have predominantly relied on RGB frames (optionally with text) while neglecting other rich sensory signals found in real-world environments. In existing designs, extending Video

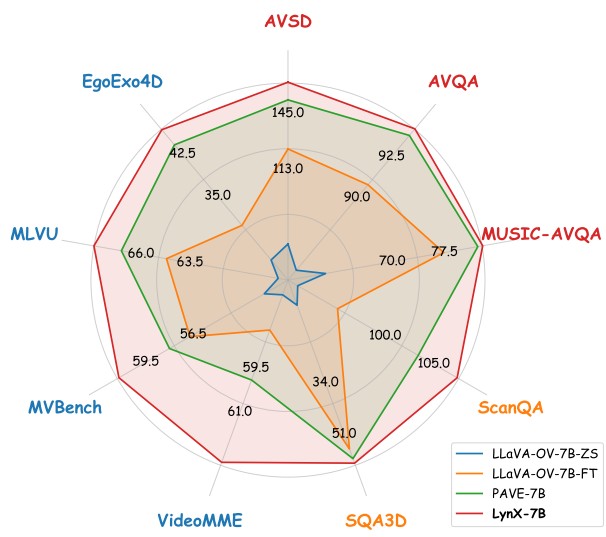

*(a)* Performance comparisons across 9 multimodal tasks

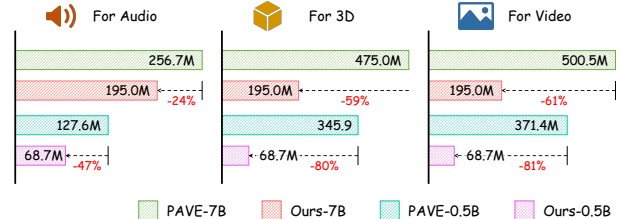

*(b)* Extra number of parameters for each new modality

*Figure 1.* V-LynX enables efficient modality expansion of pretrained Video LLMs. (a) V-LynX achieves state-of-the-art performance across diverse multimodal benchmarks with audio, 3D, and additional video, while (b) requiring significantly fewer extra parameters than PAVE (Liu et al., 2025).

LLMs to new modalities (Cheng et al., 2024; Liu et al., 2025) typically necessitates large-scale modality-specific encoders, complex fusion mechanisms, and paired supervision. Such designs significantly increase computational cost and architectural complexity, and degrade scalability.

This work investigates a fundamental question: *How can we effectively repurpose the internalized visual pathway in Video LLMs for novel modalities?* Our investigation yields a key insight that the visual encoder and projector in the Video LLM do not merely map frames onto existing vo-

[1]Yonsei University, Seoul, South Korea [2]Ewha Womans University, Seoul, South Korea. Correspondence to: Kwanghoon Sohn <khsohn@yonsei.ac.kr>, Jiyoung Lee <lee.jiyoung@ewha.ac.kr>.

*Proceedings of the 43$^{rd}$ International Conference on Machine Learning*, Seoul, South Korea. PMLR 306, 2026. Copyright 2026 by the author(s).

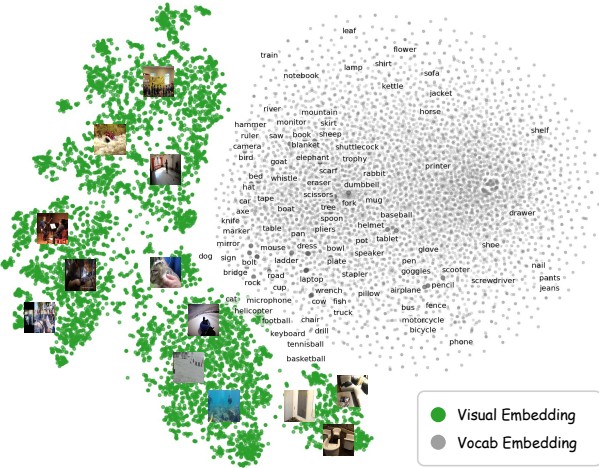

*Figure 2.* t-SNE visualization of frame embeddings and vocabulary embeddings from the pretrained LLaVA-OV (Li et al., 2025a). We randomly sample 2,000 frames from each of the six benchmarks and 10,000 token embeddings from LLaVA-OV.

cabulary embeddings. Instead, the visual pathway carves out a continuous geometric space. This emergent space, illustrated in Figure 2, functions as a bridge that decouples sensory perception from fixed vocabulary constraints, effectively allowing the LLM to process continuous visual signals as distinct, non-symbolic entities. We term such an emergent manifold as **token interface**. Like 'soft token' view in parameter-efficient prompting (Li & Liang, 2021; Lester et al., 2021), these visual tokens occupy a geometry internalized during video-language alignment training.

This perspective suggests a streamlined route for multimodal scaling: rather than retraining with a heavily connected modality encoder and projector, one needs only to map new sensory inputs into this existing token interface. Building on this, we introduce a novel token interface alignment method, **V-LynX**, that establishes a lightweight auxiliary pathway parallel to the frozen vision backbone. To ensure seamless integration, we propose a distributional alignment strategy, *i.e.* aligning both the attention responses and the statistical distributions of the new modality with the intrinsic video priors on unpaired unimodal data, for a more flexible adaptation to the target manifold without imposing over-constraints that may disrupt semantic coherence (Sun & Saenko, 2016; Gretton et al., 2012).

V-LynX shows the surprising modality expansion achievements on four new input types, including audio, 3D, high-frame-rate videos, and egocentric videos. Across all benchmarks, V-LynX consistently yields strong gains, indicating that the video interface is reliably adapted to diverse modalities. Notably, even with a compact LLaVA-OV-0.5B backbone, V-LynX outperforms PAVE (Liu et al., 2025), the prior state-of-the-art efficient multimodal alignment method,

establishing a new efficient and scalable frontier.

## 2. Related Work

**Video LLMs.** Video LLMs (Maaz et al., 2024; Li et al., 2023b) have emerged to understand and reason spatiotemporal visual instruction. Subsequent works advanced the video representation and cross-modal alignment strategy (*e.g.*, Video-LLaVA (Lin et al., 2024)), alongside efforts that scaled training data, optimization recipes, and evaluation protocols (*e.g.*, VideoChat2 (Li et al., 2024)). More recent studies such as LLaVA-OV (Li et al., 2025a), LLaVA-Video (Zhang et al., 2024), Qwen2.5-VL (Bai et al., 2025), and InternVL2.5 (Chen et al., 2024b) aim to unify image and video capabilities within a single family, and broaden task and domain coverage. In parallel, efficiency-oriented designs (Xu et al., 2024a; Weng et al., 2024) reduce tokenization overhead and computational cost for long-form video understanding. However, most works remain largely video-dominant, which limits their scalability to new modalities beyond visual inputs. In contrast, this work explores an emergent *token interface*, a connected bridge of visual and semantic representation spaces learned by Video LLMs for an efficient modality adaptation pathway.

**Video-to-multimodal LLMs.** Incorporating non-RGB signals, such as audio and 3D, into Video LLMs has recently attracted increasing research interest for richer multimodal understanding. For instance, Video-LLaMA (Zhang et al., 2023) leverages ImageBind (Girdhar et al., 2023)'s audio encoder to build a siamese audio branch to video branch, and VideoLLaMA2 (Cheng et al., 2024) strengthens audio capability by integrating a cutting-edge audio encoder (Chen et al., 2023b). While Meerkat (Chowdhury et al., 2024) targets finer spatiotemporal grounding, Video Salmonn2 (Tang et al., 2025) bootstraps language-driven audiovisual alignment by direct preference optimisation (DPO) (Rafailov et al., 2023). Instruction-tuned 3D LLMs typically rely on dedicated 3D encoders and paired supervision (Xu et al., 2024b; Chen et al., 2024a). PAVE (Liu et al., 2025) represented the closest line of work to ours in that it augments Video LLMs with an external encoder and cross-attention block-based alignment trained on multimodal pairing. Our interesting ideas rely on reusing the video pathway to LLM to adapt the distribution of new modality inputs into the video-induced token interface using only unimodal data and minimal learnable parameters.

## 3. Method

A standard Video LLM architecture typically comprises a vision encoder $g_\psi$, a projector module $p_\theta$, and an LLM $f_\phi$. Given a video $\mathbf{X}_v$, the visual pathway generates a sequence

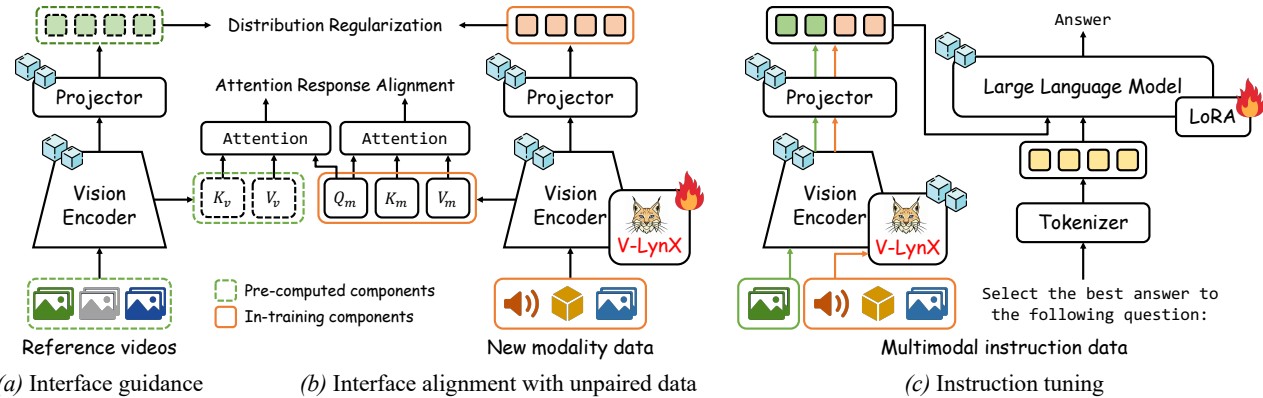

*(a)* Interface guidance  ·  *(b)* Interface alignment with unpaired data  ·  *(c)* Instruction tuning

*Figure 3.* Overall framework of our V-LynX. (a) We first extract interface guidance from a set of available videos and (b) learn LoRAs in the vision encoder to adapt the interface to given new modality data through attention response alignment and distribution regularization. (c) We then train additional LoRAs in the LLM on diverse instruction datasets.

of latent tokens $\mathbf{Z}_v$ that the LLM can interpret:

$$\mathbf{Z}_v = p_\theta(g_\psi(\mathbf{X}_v)). \tag{1}$$

These visual tokens $\mathbf{Z}_v$ inhabit a functional token interface within the LLM's high-dimensional space. The existence of this interface suggests that the pretrained visual pathway $(\theta, \psi)$ has already internalized a geometric prior for LLM-compatible sensory tokens. Consequently, extending the model to a new modality $\mathbf{X}_m$ does not necessitate a complete architectural overhaul or paired multimodal supervision. Instead, a key challenge is to make tokens from the new modality compatible with the model's native video behavior, including the attention responses inside the encoder and the token distribution expected by the projector and LLM. To this end, our V-LynX repurposes the frozen visual backbone to accommodate novel sensory inputs, where the distributional alignment preserves the attention dynamics and statistical properties internalized during video-language training. The overall procedure of V-LynX is shown in Figure 3 and Algorithm 1.

### 3.1. Shared video-path for novel modality

Integrating new modality in Video LLMs is fundamentally constrained by two factors: the dependence on paired cross-modal datasets (Akbari et al., 2021) and the risk of catastrophic forgetting when reusing or adapting existing encoders (Zhou et al., 2025a). While paired data enables explicit alignment, it is costly and inflexible, and encoder adaptation often compromises previously acquired knowledge. To overcome these limitations, V-LynX reuses the frozen vision encoder with a small set of learnable parameters $\Delta\psi$, implemented via low-rank adaptation modules (*i.e.*, LoRA (Hu et al., 2022)) within self-attention layers. Namely, the visual pathway by routing visual tokens through the frozen parameters $\psi$ during inference, while selectively activating additional learnable parameters $\Delta\psi$ only for the

new modality. By reusing the same architectural path for both modalities, the model supports efficient modality extension without a separate interface (Liu et al., 2025).

### 3.2. Interface alignment with unpaired unimodal data

While the shared-path architecture ensures parameter-efficient adaptation, effective integration of a new modality further requires aligning its representations with the token interface expected by the LLM. Conventional alignments (Akbari et al., 2021; Girdhar et al., 2023) rely on paired cross-modal supervision (*e.g.*, 3D-video-text, audio-video-text), which is often prohibitively scarce or unavailable for diverse target modalities. To circumvent these constraints, V-LynX learns additional parameters solely on unimodal data (*e.g.*, audio, 3D, and multi-view).

**Video-derived interface guidance.** To anchor modality adaptation to the interface expected by the LLM, the behavior of the pretrained Video LLM is first characterized on its native modality (*i.e.*, videos). Specifically, a set of available unlabeled videos, $\mathcal{V}$, is used to estimate reference statistics that describe how visual tokens are processed within the encoder and subsequently projected to the LLM's token space. At the encoder level, we extract averaged Key and Value embeddings at each attention layer:

$$\begin{aligned} K_v^{(l)} &= \mathbb{E}_{\mathbf{X}_v \sim \mathcal{V}}[K_\psi^{(l)}(\mathbf{X}_v)], \\ V_v^{(l)} &= \mathbb{E}_{\mathbf{X}_v \sim \mathcal{V}}[V_\psi^{(l)}(\mathbf{X}_v)], \end{aligned} \tag{2}$$

where $\mathbf{X}_v$ is an input video sampled from $\mathcal{V}$, $K_\psi^{(l)}$ and $V_\psi^{(l)}$ are projections producing Key and Value at the $l$-th layer, respectively. These mean embeddings capture the typical attention space induced by videos and serve as stable anchors for encoder-level alignments. Simultaneously, the distribution of latent video tokens is characterized at the projector level. Let $\mathbf{Z}_v = p_\theta(g_\psi(\mathbf{X}_v))$ denote the projector

output. The mean $\mu_v$ and variance $\sigma_v^2$ of the projected video embeddings are computed as

$$\mu_v = \mathbb{E}_{\mathbf{X}_v \sim \mathcal{V}}[\mathbf{Z}_v], \quad \sigma_v^2 = \mathbb{E}_{\mathbf{X}_v \sim \mathcal{V}}[(\mathbf{Z}_v - \mu_v)^2]. \quad (3)$$

The pre-computed reference serves as the target statistic to enable new modality tokens to be compatible with LLM.

**Attention response alignment.** The proposed attention alignment objective is introduced to ensure that inputs from a new modality activate the shared visual pathway in a manner compatible with existing video priors. In Video LLMs, the vision encoder constitutes the earliest stage at which heterogeneous inputs are processed through a common computational structure, and its internal attention dynamics largely determine how information is selected, aggregated, and propagated to downstream modules. We insist that while the projector and LLM operate on encoder outputs, they work on summarized token-level reasoning (Li et al., 2025b). For effective modality integration, alignment must therefore be applied at the level of encoder attention, where functional computation is formed.

Given a new modality input $\mathbf{X}_m$ of a set of newly introduced target modality data $\mathcal{M}$, Query, Key, and Value embeddings are obtained from the encoder with original and learnable parameters (*i.e.*, $\psi + \Delta\psi$) at each layer. The target attention response $O_m^{(l)}$ of $\mathbf{X}_m$ is,

$$O_m^{(l)} = \text{Attn}(Q_m^{(l)}, K_m^{(l)}, V_m^{(l)}). \quad (4)$$

The reference response $\tilde{O}_m^{(l)}$ is computed via video-derived Key $K_v^{(l)}$ and Value $V_v^{(l)}$ as references, providing a stable and well-calibrated attention behavior:

$$\tilde{O}_m^{(l)} = \text{Attn}(Q_m^{(l)}, K_v^{(l)}, V_v^{(l)}). \quad (5)$$

The Key-Value embeddings define how tokens are matched and aggregated within the shared attention framework, which directly shapes the functional operation of the encoder. By conditioning on the same Query embedding $Q_m^{(l)}$ while replacing the Key-Value pairs with video, the reference response specifies how the new modality should interact with the existing attention mechanism to remain compatible with the video-derived interface. The attention alignment loss minimizes the discrepancy between the target and reference attention responses:

$$\mathcal{L}_{\text{attn}} = \sum_l ||O_m^{(l)} - \tilde{O}_m^{(l)}||_1. \quad (6)$$

This objective promotes internal cross-modal alignment rather than raw feature similarity. Therefore, pair-independent modality adaptation is achieved while preserving the original vision-language interface.

**Distribution regularization.** Attention alignment alone does not guarantee that the projector's outputs lie in the distribution the LLM expects. To constrain the distribution (*i.e.*,

mean and variance) of new modality, we compute a statistic of projected modality tokens $\mathbf{Z}_m = p_\theta(g_{\psi+\Delta\psi}(\tilde{x}_m))$ obtained by the projector $p_\theta$:

$$\mu_m = \mathbb{E}_{\mathbf{X}_m \sim \mathcal{M}}[\mathbf{Z}_m], \quad \sigma_m^2 = \mathbb{E}_{\mathbf{X}_m \sim \mathcal{M}}[(\mathbf{Z}_m - \mu_m)^2]. \quad (7)$$

We align the token distributions by applying the mean-squared error between a reference distribution $\mathbf{Z}_v$ and the learned distribution $\mathbf{Z}_m$:

$$\mathcal{L}_{\text{stat}} = ||\mu_v - \mu_m||_2 + ||\sigma_v^2 - \sigma_m^2||_2. \quad (8)$$

**Overall objective.** We train the LoRA parameters $\Delta\psi$ in the encoder with the following objective:

$$\mathcal{L}_{\text{V-LynX}} = \mathcal{L}_{\text{attn}} + \beta \cdot \mathcal{L}_{\text{stat}}, \quad (9)$$

where $\beta$ controls the trade-off between attention alignment and training stability. Given that we do not require an additional modality-specific encoder and paired multimodal data, our V-LynX is data- and parameter-efficient solution to establish multimodal LLMs.

### 3.3. Instruction tuning

After alignment learning, $p_\theta(g_{\psi+\Delta\psi}(\cdot))$ produces the tokens from new modality data in a form that the LLM can interpret. Conditioned on the embeddings from visual and new modality data (*i.e.*, $\mathbf{Z}_v$ and $\mathbf{Z}_m$), we perform supervised fine-tuning by applying additional LoRA layers to the LLM. Specifically, we train a set of LoRA parameters $\Delta\phi$ to enable the LLM to maximize the likelihood of the autoregressively generated answer:

$$\mathcal{L}_{\text{sft}} = -\sum_{n=1}^{N} \log P(\mathbf{a}_n|\mathbf{A}_{<n}, \mathbf{Q}, \mathbf{Z}_v, \mathbf{Z}_m), \quad (10)$$

where $N$ is the number of tokens in the answer, $\mathbf{A}_{<n} = \{\mathbf{a}_1, ..., \mathbf{a}_{n-1}\}$ denotes the sequence of tokens prior to the autoregressive decoding step $n$, and $\mathbf{Q}$ is a set of instruction tokens.

### 3.4. Interpretation of V-LynX.

Recently, Huh et al. (2024) suggests that representations learned across different models and modalities may share structural regularities of the underlying world. From the Platonic representation perspective (Huh et al., 2024), the token interface can be interpreted as the organized structural regularity for the LLM. We speculate that V-LynX's formulation works on such an interpretation: if a new modality contains a world structure that overlaps with video, it can be adapted by learning a modality-specific pathway into this existing interface. Accordingly, V-LynX aligns new modality inputs to the attention behavior and projector-level token

*Table 1.* Performance comparison on audio-visual QA. We report CIDEr score on AVSD and the accuracy (Acc.) on AVQA and MUSIC-AVQA, respectively. 'ΔParams.' indicates the number of additional parameters than LLaVA-OV-0.5B/-7B.

| Method | AVSD | AVQA | MUSIC-AVQA | | | | ΔParams. |
|---|---|---|---|---|---|---|---|
| | CIDEr | Acc. | Audio Acc. | Visual Acc. | Audio-Visual Acc. | Overall Acc. | |
| *Zero-shot Video LLMs* | | | | | | | |
| CAT-7B (Ye et al., 2024) | 79.0 | - | - | - | - | 48.6 | - |
| LLaVA-OV-0.5B (Li et al., 2025a) | 65.1 | 77.4 | 60.0 | 57.1 | 48.5 | 52.8 | - |
| LLaVA-OV-7B (Li et al., 2025a) | 70.6 | 85.6 | 68.8 | 70.6 | 52.8 | 60.4 | - |
| *Task-specific models < 7B* | | | | | | | |
| COST (Pham et al., 2022) | 108.5 | - | - | - | - | - | - |
| PSTP-Net (Li et al., 2023a) | - | 90.2 | - | - | - | - | - |
| VAST (Chen et al., 2023a) | - | - | - | - | - | 80.7 | - |
| LLaVA-OV-0.5B-FT (Li et al., 2025a) | 117.6 | 86.4 | 69.6 | 76.3 | 62.8 | 67.6 | 35.2M |
| PAVE-0.5B (Liu et al., 2025) | 134.5 | 90.4 | 77.3 | 89.8 | 74.1 | 78.8 | 127.6M |
| V-LynX-0.5B (**Ours**) | **145.7** | **93.1** | **78.9** | **92.2** | **76.5** | **81.1** | 68.7M |
| *Task-specific models ≥ 7B* | | | | | | | |
| CAT-7B-FT (Ye et al., 2024) | - | 92.0 | **84.9** | 86.1 | **83.2** | **84.3** | - |
| LLaVA-OV-7B-FT (Li et al., 2025a) | 124.9 | 90.8 | 75.4 | 89.3 | 72.3 | 77.4 | 161.5M |
| PAVE-7B (Liu et al., 2025) | 152.9 | 93.8 | 79.7 | 93.0 | 78.0 | 82.3 | 256.7M |
| V-LynX-7B (**Ours**) | **163.0** | **94.2** | 80.8 | **93.5** | 78.8 | 83.0 | 195.0M |

statistics of the pretrained token interface, enabling the LLM to interpret them through the same operational regime while preserving the original video-language pathway. We provide more analysis in Section B.1.

# 4. Experiment

## 4.1. Default configuration

**Video LLM backbone.** LLaVA-OneVision (LLaVA-OV) (Li et al., 2025a) is our base Video LLM. LLaVA-OV employs SigLIP (Zhai et al., 2023) as the vision encoder $g_\psi$, Qwen-2 (Team et al., 2024) as the LLM $f_\phi$, and a 2-layer MLP as the projector $p_\theta$. To validate the scalability of our V-LynX, we primarily employ LLaVA-OV-0.5B and -7B.

**Reference videos.** For a video-derived interface guidance, we first gather a set of reference videos $\mathcal{V}$ from training sets of all benchmarks, including AVSD (Alamri et al., 2019), AVQA (Yang et al., 2022), MUSIC-AVQA (Li et al., 2022), ScanNet (Dai et al., 2017), a subset of LLaVA-Video-178K (Zhang et al., 2024), and Ego-Exo4D (Grauman et al., 2024), with a total number of videos of ∼117k. Given $\mathcal{V}$, we sample frames with 1 fps and extract Key, Value, and video token distribution are computed across all video features from the pretrained vision encoder and projector. In this regard, we will explore the effectiveness according to the scale of $\mathcal{V}$ in Table 7.

**LoRAs in $\Delta\psi$ and $\Delta\phi$.** We set rank $r$ to 64 for LoRAs in both the vision encoder ($\Delta\psi$) and LLM ($\Delta\phi$). Meanwhile, $\alpha$ is set to 128 and 16 for $\Delta\psi$ and $\Delta\phi$, respectively. We keep the identical settings across the modality.

**Baselines.** We mainly compare our V-LynX with two ap-

proaches: (1) LLaVA-OV-FT, fine-tuned through instruction tuning by LoRA without target modality data; and (2) PAVE (Liu et al., 2025), which employs a target modality encoder and incorporates it via the cross-attention module. They share the same Video LLM backbone, allowing for a fair comparison. In addition, we provide zero-shot performance of LLaVA-OV-0.5B, and -7B.

## 4.2. Audio-visual QA

Audio, while inherently synchronized with video in nature, remains a structurally heterogeneous modality that presents a significant representation gap for vision-centric models. We extend Video LLMs by assessing integrated sensory reasoning through audio-visual QA.

**Datasets.** We evaluate our V-LynX on three audio-visual QA benchmarks: **AVSD** (Alamri et al., 2019) consists of 79k open-ended QA pairs with 7.9k videos for training and 1k audio-visual questions for evaluation. We report the CIDEr score. **AVQA** (Yang et al., 2022) contains 40k videos with each closed-form QA pair for training. We evaluate with 17k questions and report the accuracy. **MUSIC-AVQA** (Li et al., 2022) provides questions, which are categorized into visual, audio, and audio-visual questions. We use 32k QA pairs from 9.2k videos for training and measure the accuracy on 9.1k questions.

**Preprocessing.** We resample the audio to 16 kHz and convert waveforms into normalized log-mel spectrograms.

**Additional baselines.** We consider task-specific models that are fine-tuned on the target dataset, including COST (Pham et al., 2022), PSTP-Net (Li et al., 2023a), CAT-7B (Ye et al., 2024), and VAST (Chen et al., 2023a); and zero-shot

*Table 2.* 3D reasoning on 3D QA benchmarks. We report CIDEr, BLEU-4, METEOR, ROUGE scores for ScanQA, and top-1 Exact Match (EM@1) and (refined EM@1) for both ScanQA and SQA3D, respectively.

| Method | ScanQA | | | | | SQA3D | ΔParams. |
| --- | --- | --- | --- | --- | --- | --- | --- |
| | CIDEr | BLEU-4 | METEOR | ROUGE | EM@1 | EM@1 | |
| *Zero-shot Video LLMs* | | | | | | | |
| VideoChat2-7B (Li et al., 2024) | 49.2 | 9.6 | 9.5 | 28.2 | 19.2 | 37.3 | - |
| LLaVA-OV-0.5B (Li et al., 2025a) | 17.2 | 1.2 | 13.7 | 18.4 | 0.2 (28.0) | 0.8 (43.0) | - |
| LLaVA-OV-7B (Li et al., 2025a) | 91.0 | 5.3 | 18.2 | 45.9 | 26.7 | 8.3 | - |
| *Task-specific models < 7B* | | | | | | | |
| LLaVA-OV-0.5B-FT (Li et al., 2025a) | 70.5 | 6.5 | 14.3 | 36.9 | 20.1 (36.3) | 44.1 (45.7) | 35.2M |
| PAVE-0.5B (Liu et al., 2025) | 84.2 | 13.1 | 17.0 | 42.1 | 23.1 (40.0) | 51.1 (52.8) | 345.9M |
| V-LynX-0.5B (**Ours**) | **87.1** | **14.3** | **17.2** | **43.8** | **26.4 (44.2)** | **52.2 (54.2)** | 68.7M |
| *Task-specific models ≥ 7B* | | | | | | | |
| 3D-LLM-7B (Hong et al., 2023) | 74.5 | 12.9 | 15.1 | 37.5 | 21.2 | 49.8 | - |
| LEO-7B (Huang et al., 2024) | 101.4 | 13.2 | 20.0 | 49.2 | 24.5 (47.6) | 50.0 (52.4) | - |
| Scene-LLM-7B (Fu et al., 2025b) | 80.0 | 12.0 | 16.6 | 40.0 | 27.2 | 54.2 | - |
| LLaVA-3D-7B (Zhu et al., 2025) | 91.7 | 14.5 | 20.7 | 50.1 | 27.0 (45.0) | 55.6 (57.6) | - |
| LLaVA-OV-7B-FT (Li et al., 2025a) | 95.1 | 13.5 | 19.1 | 47.4 | 27.4 (46.3) | 55.8 (58.1) | 161.5M |
| PAVE-7B (Liu et al., 2025) | 103.4 | 16.0 | 19.9 | 49.0 | 29.1 (48.5) | 59.0 (61.4) | 475.0M |
| V-LynX-7B (**Ours**) | **107.4** | **16.7** | **20.8** | **50.3** | **29.7 (48.6)** | **60.5 (62.6)** | 195.0M |

performance from CAT-7B.

**Results.** Table 1 shows audio-visual reasoning performance evaluated on three benchmarks. Our V-LynX consistently improves over both zero-shot and task-specific baselines, indicating that the video-induced token interface can be reliably transferred to audio. Notably, V-LynX-0.5B outperforms LLaVA-OV-0.5B-FT and PAVE-0.5B across all benchmarks. Comparison between V-LynX-7B and PAVE-7B further highlights the effectiveness of V-LynX, improving AVSD by +10.1 CIDEr and MUSIC-AVQA by +0.7% with 24% fewer additional parameters. Even though CAT-7B-FT was tailored to audio-visual LLM with aligned audio and video encoders (Girdhar et al., 2023), V-LynX-7B achieves the best score in AVQA.

### 4.3. 3D QA

We now consider 3D information as new modality data and evaluate the model on 3D QA tasks. The goal of 3D QA is to answer questions about the objects in a 3D scene and their relationships, such as relative spatial positions.

**Datasets.** We evaluate the 3D QA performance on two benchmarks that share the same 3D scanning dataset, *i.e.*, ScanNet (Dai et al., 2017). Since the following two benchmarks share most of the videos, the vision encoder is trained to share, and LoRAs in LLM are separately learned in instruction tuning stages. **ScanQA** (Azuma et al., 2022) contains 25k QA pairs for training and 4.6k questions for evaluation. We report the CIDEr, BLEU-4, METEOR, ROUGE, and top-1 Exact Match (EM@1) scores. **SQA3D** (Ma et al., 2023) includes 26k QA pairs and 3.5k questions for training and evaluation, respectively. We report the EM@1 score.

**Preprocessing.** Following (Girdhar et al., 2022), the depth map is converted into disparity maps, then processed as a 3-channel image with an RGB LookUp Table. Different from the previous works (Zhu et al., 2025; Liu et al., 2025), which take geometry-aggregated multi-view features, our V-LynX requires only a depth map for 3D QA.

**Additional baselines.** As baselines, we consider finetuned 3D-specific models, such as 3D-LLM (Hong et al., 2023), LEO (Huang et al., 2024), Scene-LLM (Fu et al., 2025b), and LLaVA-3D (Zhu et al., 2025), and the zero-shot performance from VideoChat2 (Li et al., 2024).

**Results.** As shown in Table 2, V-LynX-0.5B attains the strongest performance among sub-7B baselines, achieving 87.1 CIDEr and 26.4 EM@1 on ScanQA, and 52.2 EM@1 on SQA3D while introducing only 68.7M additional parameters. Scaling to 7B further improves accuracy across the baselines: V-LynX-7B delivers the best overall results, reaching 107.4 CIDEr and 29.7 EM@1 on ScanQA, and 60.5 EM@1 on SQA3D. Notably, it surpasses PAVE-7B (Liu et al., 2025) while requiring 59% fewer added parameters (195.0M vs. 475.0M), and consistently outperforms LLaVA-OV-7B-FT (Li et al., 2025a) (161.5M) with only a modest increase in adaptation cost. Collectively, these results indicate that V-LynX sustains the benefits of scaling while keeping modality adaptation lightweight, outperforming heavier patching-based alternatives at both model sizes. In addition, it is worth noting that our V-LynX outperforms baselines without geometry-aggregated multi-view features in (Zhu et al., 2025; Liu et al., 2025). We provide additional results evaluated on SQA3D with different backbones, including Qwen2.5-VL-3B (Bai et al., 2025) and InternVL-2.5-4B (Chen et al., 2024b) in Section B.2.

*Table 3.* High-frame-rate video understanding on diverse video QA benchmarks. Accuracy scores are reported. For MVBench, we report the performance evaluated on stage change (SC), fine-grained pose (FGP), and object shuffle (OS) subsets, following Liu et al. (2025).

| Method | VideoMME | | | | MVBench | | | | MLVU | ΔParams. |
|---|---|---|---|---|---|---|---|---|---|---|
| | Short | Median | Long | Avg. | SC | FGP | OS | Avg. | Acc. | |
| *Task-specific models < 7B* | | | | | | | | | | |
| LLaVA-OV-0.5B (Li et al., 2025a) | 53.4 | 41.2 | 37.3 | 44.0 | 37.5 | 49.0 | 33.0 | 45.5 | 50.3 | - |
| PAVE-0.5B (Liu et al., 2025) | 57.8 | 42.7 | 37.4 | 46.0 | 40.0 | 54.0 | 35.5 | 46.6 | 51.6 | 371.4M |
| V-LynX-0.5B (**Ours**) | **63.1** | **50.7** | **44.6** | **52.8** | **45.5** | **54.5** | **37.5** | **53.7** | **55.0** | 68.7M |
| *Task-specific models ≥ 7B* | | | | | | | | | | |
| LLaVA-OV-7B (Li et al., 2025a) | 70.1 | 56.6 | 48.9 | 58.2 | 52.0 | 53.0 | 35.5 | 56.7 | 64.7 | - |
| PAVE-7B (Liu et al., 2025) | 71.1 | 59.4 | 49.2 | 59.9 | 51.5 | **54.5** | 39.0 | 58.0 | 67.0 | 500.5M |
| V-LynX-7B (**Ours**) | **73.0** | **61.2** | **53.8** | **62.7** | **53.5** | 54.0 | **42.0** | **61.2** | **68.4** | 195.0M |

## 4.4. Enhanced video QA

In video understanding (VU), high-frame-rate video offers fine-grained motion cues and short-lived temporal dynamics, capturing fast actions and subtle interactions (Feichtenhofer et al., 2019; Park et al., 2023). We rethink such videos as additional information and evaluate the model on multi-domain generalized VU tasks.

**Datasets.** Following Liu et al. (2025), we train the model only on LLaVA-Video-178K and report the accuracy on remaining evaluation benchmarks. **LLaVA-Video-178K** (Zhang et al., 2024) is a large-scale video instruction-tuning dataset. In line with (Liu et al., 2025), we train on a 114k QA subset, consisting of 57k videos (each > 1 min) with two QA pairs per video. **VideoMME** (Fu et al., 2025a) provides a comprehensive multi-domain video QA benchmark with 900 videos and 2.7k four-option multiple-choice questions. **MVBench** (Li et al., 2024) contains 20 VU subtasks (*e.g.*, object shuffle and fine-grained pose), with about 3.9k videos and 4k questions in total. **MLVU** (Zhou et al., 2025b) focuses on long-video understanding and includes 1.3k long videos and 2.1k questions.

**Preprocessing.** Processing high-frame-rate videos inevitably incurs more computational overhead. To mitigate this, we adopt a frame-stacking strategy (Park et al., 2023) that aggregates temporal information into a single spatial representation. We downsample the spatial dimensions of four consecutive frames by a factor of 0.5 and tile them into a single frame. Consequently, the vision encoder processes ×4 temporal context with the same computational cost.

**Results.** Table 3 demonstrates that V-LynX-0.5B delivers the best performance across all benchmarks while remaining markedly parameter-efficient. Our V-LynX outperforms PAVE by +6.8%, +7.1%, and +3.4% on VideoMME, MVBench, and MLVU, respectively, despite introducing 81% fewer additional parameters. Moreover, V-LynX-7B attains the top overall accuracy, *i.e.*, 62.7 on VideoMME, 61.2 on MVBench, and 68.4 on MLVU. The only exception is MVBench fine-grained pose (FGP), where V-LynX under-

*Table 4.* Multi-view video understanding on DPE benchmark.

| Method | Acc. | ΔParams. |
|---|---|---|
| *Zero-shot Video LLMs* | | |
| LLaVA-OV-0.5B (Li et al., 2025a) | 23.6 | - |
| LLaVA-OV-7B (Li et al., 2025a) | 23.6 | - |
| *Task-specific models* | | |
| LLaVA-OV-0.5B-FT (Li et al., 2025a) | 28.2 | 35.2M |
| LLaVA-OV-7B-FT (Li et al., 2025a) | 29.8 | 161.5M |
| TimeSFormer (Bertasius et al., 2021) | 43.7 | - |
| PAVE-0.5B (Liu et al., 2025) | 32.4 | 41.4M |
| PAVE-7B (Liu et al., 2025) | 44.2 | 170.5M |
| V-LynX-0.5B (**Ours**) | 38.6 | 68.7M |
| V-LynX-7B (**Ours**) | **46.9** | 195.0M |

performs by 0.5%; we attribute this to resolution-sensitive cues being attenuated by the downsampling used in preprocessing. Furthermore, while V-LynX maintains a minimal parameter footprint, PAVE's reliance on modality-specific backbones–such as 330M for video encoder from Zhu et al. (2023)–leads to a prohibitive parameter explosion as the number of supported modalities increases.

## 4.5. Multi-view video understanding

Due to distinct FoV and motion patterns (Grauman et al., 2024; Park et al., 2025), we treat egocentric (first-person) videos as a separate modality. Since they naturally match the vision encoders input format, we process them directly.

**Datasets.** Following Liu et al. (2025), we employ demonstrator proficiency estimation (**DPE**) benchmark from Ego-Exo4D (Grauman et al., 2024) that aims to classify human action proficiency into one of four skill levels from a time-synchronized multi-view videos (one ego video and optionally four exo videos). We report accuracy scores.

**Additional baselines.** We include TimeSFormer (Bertasius et al., 2021) trained on paired ego-exo videos.

**Results.** In Table 4, the zero-shot performance of the baselines suggests that, despite strong video-language priors,

*Table 5.* Component analysis in V-LynX on ScanQA.

| Method | C | B-4 | M | R | EM@1 |
|---|---|---|---|---|---|
| V-LynX | 87.1 | 14.3 | 17.2 | 43.8 | 26.4 (44.2) |
| – Attn. Align. | 81.0 | 11.8 | 16.3 | 41.2 | 23.5 (40.4) |
| – Dist. Reg. | 86.2 | 13.4 | 17.1 | 43.5 | 25.6 (43.0) |
| – Interface Adapt. | 77.3 | 10.9 | 15.7 | 39.1 | 22.4 (39.9) |

*Table 6.* Different rank $r$ of LoRAs in $\Delta\psi$ on ScanQA.

| $r$ | C | B-4 | M | R | EM@1 | $\Delta$Params. |
|---|---|---|---|---|---|---|
| 8 | 86.1 | 13.1 | 16.2 | 42.8 | 24.9 (42.7) | 39.4M |
| 16 | 86.8 | 13.6 | 17.1 | 43.0 | 25.3 (43.3) | 43.6M |
| 32 | 86.9 | 13.7 | 17.2 | 43.3 | 26.3 (44.0) | 51.9M |
| 64 | 87.1 | 14.3 | 17.2 | 43.8 | 26.4 (44.2) | 68.7M |

*Table 7.* Reference $\mathcal{V}$ choices on ScanQA. $|\mathcal{V}|$ indicate the number of videos.

| Source | C | B-4 | M | R | EM@1 | $|\mathcal{V}|$ |
|---|---|---|---|---|---|---|
| Audio | 87.7 | 14.2 | 17.3 | 43.8 | 25.9 (43.6) | 57k |
| 3D | 87.8 | 14.8 | 17.3 | 43.7 | 26.3 (43.9) | 563 |
| Video | 87.8 | 14.3 | 17.2 | 43.7 | 25.8 (43.5) | 59k |
| All | 87.1 | 14.3 | 17.2 | 43.8 | 26.4 (44.2) | 117k |

pretrained LLMs lack the egocentric-specific cues required for reliable proficiency estimation. Comparisons between task-specific models show that our V-LynX consistently outperforms TimeSFormer and PAVE. While V-LynX introduces a marginal parameter increment to capture the domain shift in egocentric videos, unlike PAVE, which relies on a shared encoder for disparate perspectives, our V-LynX achieves remarkable improvement in both 0.5B and 7B models by +6.2% and +2.7%, respectively.

### 4.6. Ablation studies

All experiments in this section are conducted on ScanQA.

**Objective.** We compare V-LynX variants by ablating each objective used for interface alignment, *i.e.*, attention response alignment and distribution regularization. As shown in Table 5, removing attention alignment yields the larger performance drop, indicating that aligning attention responses is a primary component to adapt the pretrained model to new modality data. In contrast, ablating distribution regularization causes a minor degradation, suggesting the regularization plays a complementary role by stabilizing token statistics rather than defining the actual mapping. Finally, training LoRAs of the vision encoder and LLM through only instruction tuning (–Interface Align.) leads to a substantial collapse, demonstrating that interface alignment is essential for transferring new modality representations into the LLM-compatible token interface.

**LoRA rank in vision encoder.** Table 6 indicates that even low-rank adapters achieve strong performance. Increasing capacity achieves diminishing yet consistent improvements, where $r = 64$ provides the best results of 87.1 CIDEr and 26.4 EM@1 scores with 68.7M parameters. Overall, V-LynX is robust to the choice of $r$ and remains parameter-efficient, retaining most gains at small ranks.

**Source and scale of reference $\mathcal{V}$.** As shown in Table 7, V-LynX is remarkably resilient to distribution shifts. Using

out-of-distribution audio-related videos (57k) achieves 87.7 CIDEr and 25.9 EM@1, while a minimal, 3D set of 563 clips remains highly competitive at 87.8 CIDEr and 26.3 EM@1. Overall results underscore that V-LynX achieves accurate interface adaptation without requiring large or strictly in-domain reference $\mathcal{V}$, in that the averaged reference statistics provide a stable target across diverse sources.

### 4.7. Visualization

**Qualitative results.** We provide qualitative comparisons between our V-LynX and PAVE (Liu et al., 2025) on 3D QA and audio-visual QA in Section B.3.

**Attention visualization.** In Figure 4, we extract attention maps between question tokens and the corresponding modality tokens from a given scene-question pair from ScanQA. Especially for the 3D inputs, we illustrate the attention maps with and without our V-LynX to demonstrate the actual functionality of the modality-specific pathway. The results indicate that the model with V-LynX attends to consistent, question-relevant regions across modalities (*e.g.*, focusing on the target object areas for white pillow or monitor), suggesting that the adapted modality representations participate in the same functional routing used for reasoning.

## 5. Conclusion

In this work, we introduced the token interface, an emergent and functional manifold within Video LLMs. Leveraging this insight, we proposed V-LynX, a highly efficient framework for multimodal expansion. By aligning new modalities to this internalized interface space using only unimodal data and a lightweight auxiliary pathway, V-LynX achieves state-of-the-art performance across diverse settings, including audio-visual QA, 3D reasoning, high-frame-rate video understanding, and multi-view proficiency estimation.

**Broader impact.** V-LynX could broaden access to multimodal reasoning for data-scarce domains and resource-constrained deployments. Potential positive outcomes include improved assistive systems that jointly leverage vision with audio or geometry, and more modular multimodal pipelines that reduce redundant pretraining and associated energy costs. In addition, expanding the token-interface methodology offers a new lens through which to analyze the

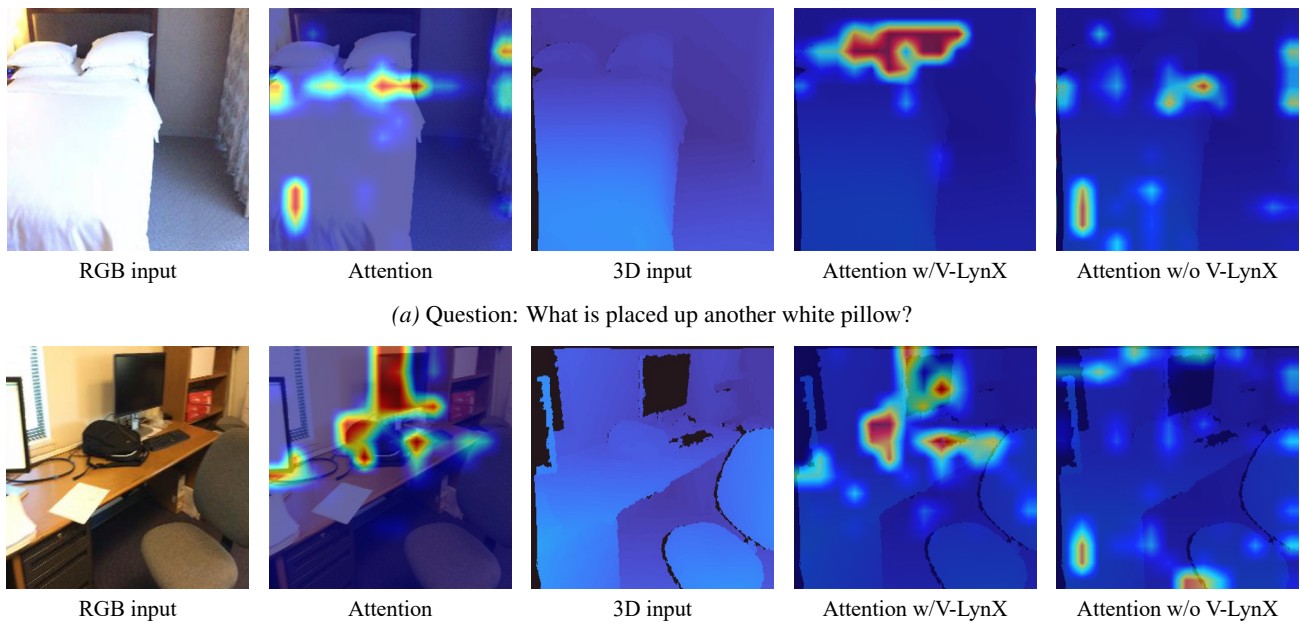

*(a)* Question: What is placed up another white pillow?

*(b)* Question: Where is the monitor with a dark screen located?

*Figure 4.* Attention visualization on ScanQA. We depict the RGB inputs and the corresponding attention maps. For the 3D inputs, we provide the attention maps with and without our V-LynX.

*modality gap* (Liang et al., 2022). By defining a measurable interface gap based on geometric statistics and functional attention divergence, researchers could perform principled diagnostics of multimodal adaptation. This framework provides a robust foundation for identifying failure modes, optimizing reference set selection, and supporting modelfairness by quantifying representation variations across domains and demographic factors.

## Impact Statement

This paper presents work whose goal is to advance multimodal machine learning by making pretrained Video LLMs more transferable to new modalities under unimodal supervision. The approach may reduce computational cost and data constraints for modality expansion, but it may also increase the potential for privacy-invasive deployments and uneven reliability across settings. We anticipate that responsible use will require careful data governance, evaluation under distribution shifts, and safeguards for sensitive audio and first-person data.

## Acknowledgements

This work was supported by the National Research Foundation of Korea(NRF) grant funded by the Korea government(MSIT) (RS-2025-16065706) and (RS-2025-02216328).

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

# A. Implementation Details

## A.1. Algorithm

Overall procedure of our V-LynX is described in Algorithm 1.

---

**Algorithm 1** V-LynX

---

**Require:** Pre-trained video LLM: Vision encoder $g_\psi$, projector $p_\theta$, LLM $f_\phi$.
**Require:** Unlabeled videos $\mathcal{V}$; unlabeled modality data $\mathcal{M}$; instruction data $\mathcal{D}$.
**Require:** Weights $\beta$.
**Ensure:** Vision encoder LoRA $\Delta\psi$; LLM LoRA $\Delta\phi$.
 1: **Reference extraction**
 2: **for** each layer $l$ in encoder **do**
 3: $\quad \bar{K}_v^{(l)} \leftarrow \mathbb{E}_{\mathbf{X}_v \sim \mathcal{V}}\big[K^{(l)}(\mathbf{X}_v)\big]$
 4: $\quad \bar{V}_v^{(l)} \leftarrow \mathbb{E}_{\mathbf{X}_v \sim \mathcal{V}}\big[V^{(l)}(\mathbf{X}_v)\big]$
 5: **end for**
 6: $\mathbf{Z}_v \leftarrow p_\theta(g_\psi(\mathbf{X}_v))$
 7: $\mu_v \leftarrow \mathbb{E}_{\mathbf{X}_v \sim \mathcal{V}}[\mathbf{Z}_v], \quad \sigma_v^2 \leftarrow \mathbb{E}_{\mathbf{X}_v \sim \mathcal{V}}\Big[(\mathbf{Z}_v - \mu_v)^2\Big]$
 8: **Stage 1: Unimodal training (optimize $\Delta\psi$)**
 9: Freeze $p_\theta$ and $f_\phi$; insert LoRA into $g_\psi$ to obtain $g_{\psi+\Delta\psi}$
10: **repeat**
11: $\quad$ Sample $\mathbf{X}_m \sim \mathcal{M}$
12: $\quad$ Run $g_{\psi+\Delta\psi}(\mathbf{X}_m)$ and collect $\{Q_m^{(l)}, K_m^{(l)}, V_m^{(l)}\}_{(l)}$
13: $\quad \mathcal{L}_{\text{attn}} \leftarrow 0$
14: $\quad$ **for** each layer $l$ in encoder **do**
15: $\quad\quad O_m^{(l)} \leftarrow \text{Attn}\Big(Q_m^{(l)}, K_m^{(l)}, V_m^{(l)}\Big)$
16: $\quad\quad \tilde{O}_m^{(l)} \leftarrow \text{Attn}\Big(Q_m^{(l)}, \bar{K}_v^{(l)}, \bar{V}_v^{(l)}\Big)$
17: $\quad\quad \mathcal{L}_{\text{attn}} \leftarrow \mathcal{L}_{\text{attn}} + ||O_m^{(l)} - \tilde{O}_m^{(l)}||_1$
18: $\quad$ **end for**
19: $\quad \mathbf{Z}_m \leftarrow p_\theta(g_{\psi+\Delta\psi}(\mathbf{X}_m))$
20: $\quad \mu_m \leftarrow \mathbb{E}[\mathbf{Z}_m], \quad \sigma_m^2 \leftarrow \mathbb{E}[(\mathbf{Z}_m - \mu_m)^2]$
21: $\quad \mathcal{L}_{\text{stat}} \leftarrow ||\mu_m - \mu_v||_2 + ||\sigma_m^2 - \sigma_v^2||_2$
22: $\quad \mathcal{L}_{\text{V-LynX}} \leftarrow \mathcal{L}_{\text{attn}} + \beta\mathcal{L}_{\text{stat}}$
23: $\quad$ Update $\Delta\psi$ by minimizing $\mathcal{L}_{\text{V-LynX}}$
24: **until** convergence
25: **Stage 2: Instruction tuning (optimize $\Delta\phi$)**
26: Freeze $g_{\psi+\Delta\psi}$ and $p_\theta$; insert LoRA into $f_\phi$ to obtain $f_{\phi+\Delta\phi}$
27: **repeat**
28: $\quad$ Sample $(\mathbf{Q}, \mathbf{X}_m, \mathbf{A}) \sim \mathcal{D}$ (and optional $\mathbf{X}_v$ if available)
29: $\quad \mathbf{Z}_m \leftarrow p_\theta(g_{\psi+\Delta\psi}(\mathbf{X}_m)) \quad$ (and $\mathbf{Z}_v \leftarrow p_\theta(g_\psi(\mathbf{X}_v))$ if used)
30: $\quad \mathcal{L}_{\text{sft}} \leftarrow -\sum_{n=1}^{N} \log P(a_n \mid a_{<n}, \mathbf{Q}, \mathbf{Z}_v, \mathbf{Z}_m)$
31: $\quad$ Update $\Delta\phi$ by minimizing $\mathcal{L}_{\text{sft}}$
32: **until** convergence
33: **return** $\Delta\psi, \Delta\phi$

---

## A.2. Processing data

We transform new modality data to enable the pretrained vision encoder to process them accordingly. We provide examples for each modality data in Figure A1.

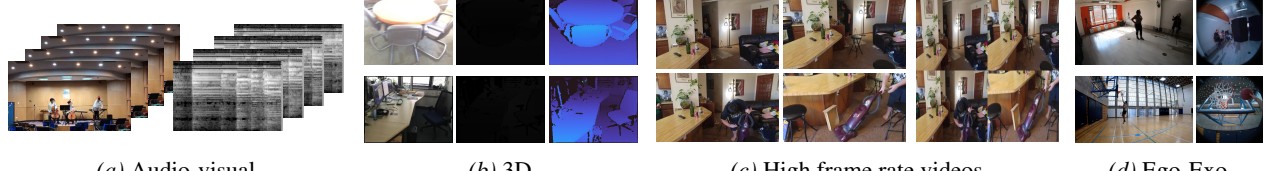

| *(a)* Audio-visual | *(b)* 3D | *(c)* High frame rate videos | *(d)* Ego-Exo |

*Figure A1.* Examples of input transformation for each new modality. (a) From a given video, we sample an audio signal and transform it into a normalized log-mel spectrogram; (b) Given a depth map, we convert it into a 3-channel disparity map; (c) We rescale and stack multiple frames to obtain a single frame. While the transformed frame has the size of the original frame, we depict it with a scaled-up size for visibility; (d) Egocentric videos are fed into the model without any transformation.

*Table B1.* Embedding analysis on the LLMs' input space. We present (1) the averaged pairwise cosine distance, (2) the L2 norm of the modality-wise mean embedding, and (3) a scale-invariant modality gap (Liang et al., 2022) between frame and vocabulary embeddings.

| Model | Cosine Distance | | | $\ell$-2 Norm | | $\Delta_{\text{gap.}}$ |
| | Vocab.-Vocab. | Frame-Frame | Vocab.-Frame | Vocab. | Frame | |
| --- | --- | --- | --- | --- | --- | --- |
| LLaVA-OV-0.5B (Li et al., 2025a) | 0.71 | 0.10 | 0.96 | 0.19 | 26.30 | 1.0081 |
| LLaVA-OV-7B (Li et al., 2025a) | 0.78 | 0.11 | 0.99 | 0.10 | 45.42 | 0.9499 |
| Qwen2.5-VL-3B (Bai et al., 2025) | 0.79 | 0.35 | 1.02 | 0.47 | 31.65 | 0.9539 |
| InternVL-2.5-4B (Chen et al., 2024b) | 0.92 | 0.11 | 1.01 | 0.29 | 41.71 | 0.9930 |

## A.3. Training details

We optimize all models with AdamW (Loshchilov & Hutter, 2019). For both interface alignment and instruction tuning, we apply a linear warm-up over the first 3% of iterations and use cosine annealing for learning rate decay (Loshchilov & Hutter, 2016). We set the regularization parameter $\beta$ to 0.01. All experiments are conducted on two NVIDIA RTX PRO 6000 Blackwell 96GB GPUs.

**Audio-visual QA.** For interface alignment, we train V-LynX for 10 epochs with batch size 8 across all audio-visual QA benchmarks, followed by instruction tuning for 1 epoch on AVSD and 2 epochs on AVQA and MUSIC-AVQA. For V-LynX-0.5B, the base learning rates are 2e-5 for interface alignment and 1e-4 for instruction tuning. For V-LynX-7B, we use 5e-4 for interface alignment and 1e-4 for instruction tuning.

**3D QA.** ScanQA and SQA3D provide 562 and 518 training videos, respectively, with 517 videos overlapping. We therefore train a single set of vision-encoder LoRAs using the ScanNet training split, and then perform instruction tuning of LLM LoRAs for 1 epoch on ScanQA and 2 epochs on SQA3D. We use the same learning rate settings as in the audio-visual QA experiments.

**Enhanced video QA.** We perform interface alignment for 5 epochs and instruction tuning for 2 epochs on LLaVA-Video-178K, and evaluate on VideoMME, MVBench, and MLVU without training on the target benchmarks. For interface alignment, we set the base learning rate to 1e-5 for V-LynX-0.5B and 5e-5 for V-LynX-7B, while keeping the remaining schedule unchanged.

**Multi-view video understanding.** We follow the same training protocol as audio-visual QA, performing 10 epochs of interface alignment and then instruction tuning for 1 epoch on AVSD and 2 epochs on AVQA and MUSIC-AVQA. We use the learning rate settings from the enhanced video QA configuration.

## B. Additional Analysis

### B.1. Analysis on token interface

**Existence of token interface.** We provide additional analysis to present that token interfaces are a common phenomenon in Video LLM. Specifically, we quantitatively analyze the LLM's input space using three statistics: (1) the averaged pairwise cosine distance, (2) the $\ell$-2 norm of the modality-wise mean embedding, and (3) a scale-invariant modality gap (Liang et al., 2022) between frame and vocabulary embeddings across four Video LLMs, including LLaVA-OV-0.5B, -7B (Li et al., 2025a), Qwen2.5-VL-3B (Bai et al., 2025), and InternVL2.5-4B (Chen et al., 2024b), as shown in Table B1. Across all backbones, projected frame embeddings show much smaller pairwise cosine distances than vocabulary embeddings,

*Table B2.* Mean and variance analysis at 26-layers of the vision tower in LLaVA-OV-0.5B (Li et al., 2025a).

| Entity | | Layers | | | | | | | | | | | | |
|---|---|---|---|---|---|---|---|---|---|---|---|---|---|---|
| | | 1 | 2 | 3 | 4 | 5 | 6 | 7 | 8 | 9 | 10 | 11 | 12 | 13 |
| Key | $\|\|K_v^{(l)}\|\|^2$ | 0.675 | 0.615 | 1.243 | 1.373 | 0.918 | 1.210 | 0.930 | 1.097 | 0.951 | 0.872 | 0.804 | 0.746 | 0.725 |
| | $\text{trace}(\Sigma_K^{(l)})$ | 0.002 | 0.003 | 0.002 | 0.002 | 0.003 | 0.003 | 0.004 | 0.003 | 0.004 | 0.005 | 0.004 | 0.006 | 0.006 |
| | $R_K$ | 369.0 | 211.7 | 816.0 | 786.2 | 267.0 | 427.1 | 263.6 | 329.8 | 219.2 | 191.2 | 181.6 | 131.3 | 114.9 |
| Value | $\|\|V_v^{(l)}\|\|^2$ | 0.015 | 0.028 | 0.056 | 0.041 | 0.164 | 0.167 | 0.108 | 0.056 | 0.074 | 0.074 | 0.054 | 0.065 | 0.052 |
| | $\text{trace}(\Sigma_V^{(l)})$ | $2\times10^{-5}$ | $3\times10^{-4}$ | $3\times10^{-4}$ | $5\times10^{-4}$ | $1\times10^{-3}$ | $1\times10^{-3}$ | $2\times10^{-3}$ | $1\times10^{-3}$ | $3\times10^{-3}$ | $2\times10^{-3}$ | $2\times10^{-3}$ | $2\times10^{-3}$ | $2\times10^{-3}$ |
| | $R_V$ | 783.0 | 92.7 | 175.2 | 81.7 | 136.6 | 132.9 | 70.7 | 43.3 | 26.1 | 43.5 | 31.8 | 30.5 | 22.1 |

| Entity | | Layers | | | | | | | | | | | | |
|---|---|---|---|---|---|---|---|---|---|---|---|---|---|---|
| | | 14 | 15 | 16 | 17 | 18 | 19 | 20 | 21 | 22 | 23 | 24 | 25 | 26 |
| Key | $\|\|K_v^{(l)}\|\|^2$ | 0.711 | 0.685 | 0.646 | 0.721 | 0.689 | 0.633 | 0.596 | 0.621 | 0.673 | 0.647 | 0.657 | 0.651 | 0.608 |
| | $\text{trace}(\Sigma_K^{(l)})$ | 0.007 | 0.007 | 0.008 | 0.008 | 0.008 | 0.008 | 0.009 | 0.010 | 0.010 | 0.010 | 0.010 | 0.010 | 0.008 |
| | $R_K$ | 108.2 | 95.1 | 85.0 | 92.3 | 83.4 | 75.3 | 68.1 | 64.9 | 67.1 | 63.2 | 63.8 | 63.0 | 74.8 |
| Value | $\|\|V_v^{(l)}\|\|^2$ | 0.053 | 0.045 | 0.035 | 0.035 | 0.040 | 0.051 | 0.048 | 0.052 | 0.093 | 0.109 | 0.158 | 0.170 | 0.293 |
| | $\text{trace}(\Sigma_V^{(l)})$ | $3\times10^{-3}$ | $4\times10^{-3}$ | $3\times10^{-3}$ | $4\times10^{-3}$ | $4\times10^{-3}$ | $6\times10^{-3}$ | $6\times10^{-3}$ | $6\times10^{-3}$ | $7\times10^{-3}$ | $8\times10^{-3}$ | $1\times10^{-2}$ | $1\times10^{-2}$ | $1\times10^{-2}$ |
| | $R_V$ | 21.3 | 12.4 | 12.0 | 9.2 | 9.3 | 8.0 | 8.3 | 8.3 | 13.6 | 13.3 | 15.4 | 16.3 | 25.2 |

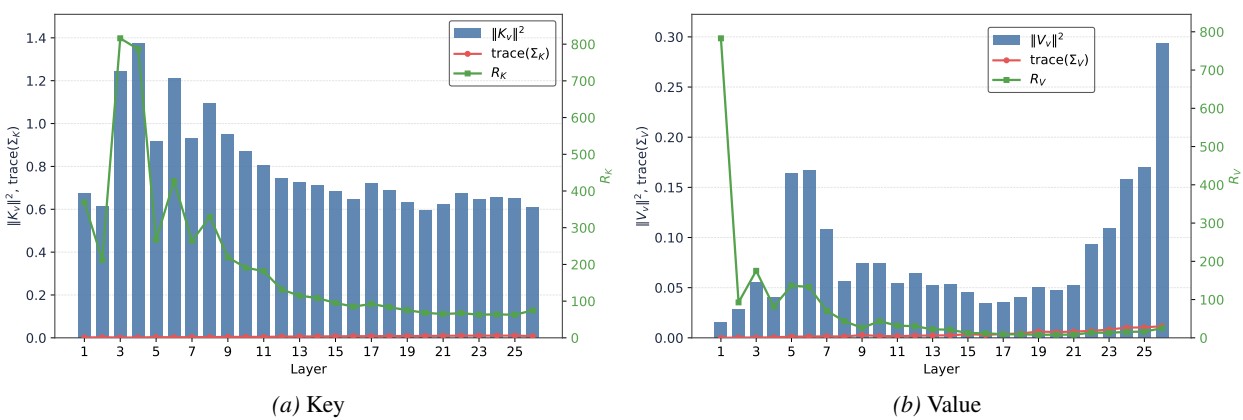

*(a)* Key  *(b)* Value

*Figure B2.* Mean, variance, and corresponding dominance score for the Key and Value of the interface guidance.

indicating that visual tokens form a more compact geometric regime. At the same time, the cosine distance between vocabulary and frame embeddings is close to one, suggesting that the two embedding groups are nearly orthogonal rather than intermingled. The consistently large scale-invariant modality gap further indicates that this separation cannot be explained by simple norm differences. Importantly, this separated region is not an invalid out-of-distribution space, but an operationally compatible token interface that can be processed by the LLMs. This interpretation is supported by our empirical results: new modalities can be aligned to this region without paired cross-modal supervision during the interface alignment stage, while removing interface alignment substantially degrades performance, as shown in Table 5.

**Video-derived interface guidance.** For interface alignment, we first estimate reference statistics by extracting averaged Key and Value embeddings at each attention layer from $\mathcal{V}$, as shown in Equation (2). While we demonstrate that they successfully represent the behavior of the pretrained Video LLM, they are potentially less representative since $\mathcal{V}$ is gathered from the six benchmarks. We further measure the variance of the Key and Value across the reference videos, and compare with the averaged Key and Value embeddings to verify the stability of the interface guidance.

Let $\mathcal{V}_s$ is the $s$-th benchmark in $\mathcal{V}$. We first obtain the mean Key and Value embeddings for each benchmark at each layer:

$$K_s^{(l)} = \mathbb{E}_{\mathbf{X}_v \sim \mathcal{V}_s} K_\psi^{(l)}(\mathbf{X}_v), \quad V_s^{(l)} = \mathbb{E}_{\mathbf{X}_v \sim \mathcal{V}_s} V_\psi^{(l)}(\mathbf{X}_v). \tag{11}$$

With the mean Key and Value embeddings, we can obtain the variance matrix $\Sigma_K$ and $\Sigma_V$:

$$\Sigma_K^{(l)} = \frac{1}{S-1} \sum_s (K_s^{(l)} - K_v^{(l)})(K_s^{(l)} - K_v^{(l)})^\top, \quad \Sigma_V^{(l)} = \frac{1}{S-1} \sum_s (V_s^{(l)} - V_v^{(l)})(V_s^{(l)} - V_v^{(l)})^\top, \tag{12}$$

where $K_v^{(l)}$, $V_v^{(l)}$ are the reference Key and Value in Equation (2) and $S$ is the number of benchmarks. We define a

*Table B3.* Performance comparisons with different backbones. We report top-1 Exact Match (EM@1) and (refined EM@1) on SQA3D.

| Method | EM@1 | ΔParams. |
|---|---|---|
| *With LLaVA-OV* | | |
| LLaVA-OV-0.5B (Li et al., 2025a) | 0.8 (43.0) | - |
| LLaVA-OV-7B (Li et al., 2025a) | 8.3 | - |
| V-LynX-0.5B (**Ours**) | 52.2 (54.2) | 68.7M |
| V-LynX-7B (**Ours**) | 60.5 (62.6) | 195.0M |
| *With Qwen2.5-VL* | | |
| Qwen2.5-VL-3B (Bai et al., 2025) | 15.1 | - |
| V-LynX-3B (**Ours**) | 59.7 (60.0) | 165.5M |
| *With InternVL-2.5* | | |
| InternVL-2.5-4B (Chen et al., 2024b) | 44.0 (50.6) | - |
| V-LynX-4B (**Ours**) | **61.1** (**63.5**) | 144.9M |

*Table B4.* Performance comparisons on AV-Human of AVUT (Yang et al., 2025). We report the accuracy (Acc.).

| Method | Acc. (%) |
|---|---|
| *Visual MLLMs* | |
| GPT-4o | 56.62 |
| Qwen2-VL-7B | 58.38 |
| LLaVA-Video-7B | 56.52 |
| InternVL2-8B | 45.9 |
| VILA-1.5-8B | 44.48 |
| VideoLLaVA-7B | 33.14 |
| *Audio MLLMs* | |
| SALMONN-13B | 36.48 |
| *Audio-visual MLLMs* | |
| Gemini 1.5 Pro | 78.34 |
| VideoLLaMA2-7B | 44.90 |
| video-SALMONN-13B | 38.33 |
| PandaGPT-13B | 25.38 |
| V-LynX-0.5B | 46.91 |

dominance score $R$ as the ratio of the magnitude of the reference Key and Value to the total variance:

$$R_K^{(l)} = \frac{||K_v^{(l)}||^2}{\text{trace}(\Sigma_K^{(l)})}, \quad R_V^{(l)} = \frac{||V_v^{(l)}||^2}{\text{trace}(\Sigma_V^{(l)})}. \tag{13}$$

In Table B2 and Figure B2, we depict the magnitude of the reference Key and Value embeddings, the total variance of the Key and Value embeddings across benchmarks, and the corresponding dominance scores derived from each layer of the vision tower in LLaVA-OV-0.5B (Li et al., 2025a). The result demonstrates that the global reference is stable: when averaged across layers, the variance is 0.006 for the Key and 0.004 for the Value, while the corresponding means are 0.80 and 0.08, respectively. Consequently, the dominance scores of the Key and Value are both much higher than 1, indicating that the reference Key and Value statistics are tightly concentrated across videos rather than being dominated by large sample-to-sample fluctuations.

### B.2. Additional experiments

**Performance with different backbones.** To further demonstrate the scalability of V-LynX in backbones, we train V-LynX with Qwen2.5-VL-3B (Bai et al., 2025) and InternVL-2.5-4B (Chen et al., 2024b), and evaluate them on SQA3D. As shown in Table B3, V-LynX consistently improves all baselines: V-LynX-0.5B and V-LynX-7B achieve 52.2 and 60.5 EM@1 with LLaVA-OV, while V-LynX-3B improves Qwen2.5-VL-3B from 15.1 to 59.7 EM@1. InternVL-2.5-4B already provides a strong baseline of 44.0 EM@1, partly because ScanQA was included in its fine-tuning data (Chen et al., 2024b). Nevertheless, V-LynX-4B further improves it to 61.1 EM@1 and 63.5 refined EM@1 with only 144.9M additional parameters. These results demonstrate that the proposed interface alignment generalizes beyond a specific Video LLM backbone.

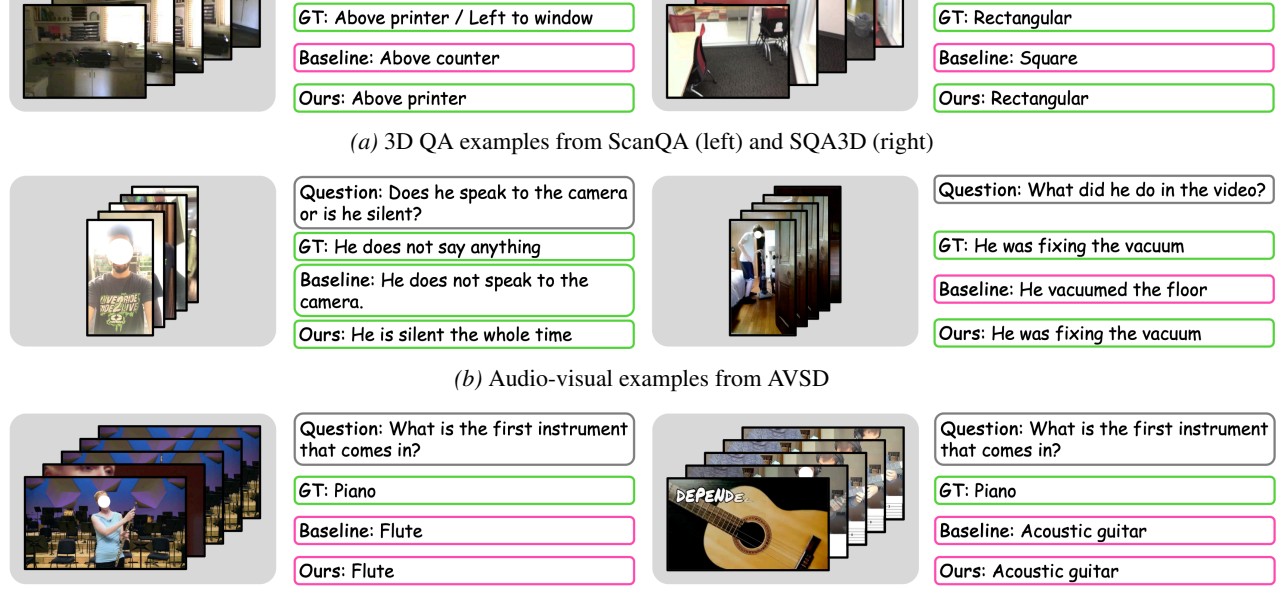

*(a)* 3D QA examples from ScanQA (left) and SQA3D (right)

*(b)* Audio-visual examples from AVSD

*(c)* Failure cases from MUSIC-AVQA

*Figure B3.* Qualitative examples for (a) 3D QA from ScanQA (left) and SQA3D (right), and (b) audio-visual QA from AVSD. We also provide (c) failure cases from MUSIC-AVQA.

**Experiment on less visually grounded task.** To further examine V-LynX on tasks where a new modality data (*i.e.*, audio) plays a central role, we conduct an additional experiment on AVUT (Yang et al., 2025). AVUT is an audio-centric video understanding benchmark designed to reduce text shortcuts and evaluate both audio content understanding and audio-visual alignment across diverse video domains. It consists of AV-Gemini, a larger Gemini-augmented training split, and AV-Human, a human-annotated evaluation split. Specifically, we train V-LynX on AV-Gemini, and evaluate it on AV-Human.

As shown in Table B4, V-LynX-0.5B achieves 46.91% accuracy on AV-Human. Although this task is less aligned with our video-induced token interface than visually grounded audio-visual QA, V-LynX still outperforms several audio and audio-visual MLLMs, including SALMONN-13B (Tang et al., 2024), VideoLLaMA2-7B (Cheng et al., 2024), video-SALMONN-13B (Sun et al., 2024), and PandaGPT-13B (Su et al., 2023). This result suggests that the proposed interface alignment can transfer audio information to the Video LLM beyond strongly visually grounded settings. At the same time, the remaining gap to Gemini 1.5 Pro and strong visual MLLMs indicates that purely audio-centric reasoning remains challenging when the new modality is routed through a video-induced interface, which is consistent with the limitation discussed in Section C.

### B.3. Qualitative analysis

In Figure B3a, the results show that V-LynX produces more spatially grounded answers than the baseline, which often defaults to course category priors and misses geometric relations (*e.g.*, relative position such as above or behind). For audio-visual QA, our V-LynX yields responses that better reflect subtle action and state cues, as shown in Figure B3b. Notably, even under spurious audio signals (*e.g.*, the vacuuming sound), V-LynX can still reach correct conclusions by appropriately integrating visual evidence with the language query. We also include failure cases from MUSIC-AVQA in Figure B3c, where both the baseline and V-LynX struggle when the target instruments are not given as visual cues (*e.g.*, piano present only as background music). Although similar limitations have been reported in (Liu et al., 2025), we attribute this to an inherent limitation of our V-LynX, which is to align a new modality to the visual token interface.

# C. Limitation

**Inherent limitation.**    Our approach adapts a new modality by aligning it to the *video-induced token interface*. This design choice bounds what the adapted modality can express to what is representable through the visual interface that the backbone has internalized. In practice, when the target concept is weakly or not at all grounded in visual evidence, alignment to the visual token interface can be insufficient. This behavior is visible in the MUSIC-AVQA failure cases where the target instrument is only present as background audio without a corresponding visual cue, leading both LynX and prior baselines to fail on purely audio-driven discrimination.

**Input transformation.** A second limitation arises from the modality-to-vision preprocessing that enables the reuse of the frozen vision encoder. While aligning heterogeneous signals into a unified visual manifold ensures seamless integration, it introduces a subtle trade-off between modality-specific granularity and cross-modal compatibility. For high-frame-rate video, the frame stacking strategy (Park et al., 2023) introduces downsampling that attenuates resolution-sensitive cues, which aligns with the observed degradation on fine-grained pose in MVBench. Similarly, depth-to-disparity conversion and audio-to-log-mel transformation may remove information that is useful for downstream reasoning, such as fine geometry, phase, or transient structure, and the model cannot recover what is lost at the interface input.

