# OpenReview forum: "V-LynX: Token Interface Alignment for Video+X LLMs"
_ICML.cc/2026/Conference — ICML 2026 regular_

### Official Review · Reviewer_HBTt · 2026-02-27

**Soundness:** 3
**Presentation:** 3
**Significance:** 3
**Originality:** 3
**Overall Recommendation:** 4
**Confidence:** 4

**Summary:**

Paper considers the question how the token-representation of video-LLMs can be repurposed for novel modalities not seen during training. The 'lynx' method is proposed that first provides reference videos to provide 'interface guidance'. Then it aligns the new modality to the video modality at the attention level using LoRAs. To assure the LLM can also handle the new modalities well a LoRA is fine-tuned there as well using multimodal instruction data.

**Compliance With Llm Reviewing Policy:**

Affirmed.

**Final Justification:**

I have appreciated the interaction with the authors. They have convincingly addressed all the identified weaknesses and I will update to weak accept.  I expect that several responses will also end up the in the updated main paper.

**Key Questions For Authors:**

Please address the weaknesses indicated above.

**Limitations:**

The paper contains no limitation section. This is another weakness.

**Strengths And Weaknesses:**

Strengths:
* How to levarage multimodal sensory data beyond the ones seen during training is an intriguing topic that warrants more community interest.

* Experimental results are strong and extensive.

* Code is provided.

Weaknesses:

* The paper presents an integrated system that works well, but there are limited lessons to be learned from the work and leaves me with a lot of questions, e.g.
   * There is no clear novelty nor contribution statement, what does this paper contribute?
   * The method description is not very clear, and lacks theoretical grounding. Is there a relationship with the Platonic representation hypothesis?
   * Learning modality alignment without paired cros-modal supervision is not novel, see for example:
Learning Unseen Modality Interaction,
Yunhua Zhang, Hazel Doughty, Cees G.M. Snoek,
arXiv:2306.12795

   * I don't understand the 'interface guidance' process and how sensitive the approach is to this part of the pipeline. The videos it sees before alginment are all taken from the same datasets used during evaluation. Table 7 indicates hardly any difference between datasets used, so what is the purpose of this step?


* Minor: The name 'Lynx' is not very original, there are already many papers using the same name:
   * Lynx: Towards High-Fidelity Personalized Video Generation,
Shen Sang, Tiancheng Zhi, Tianpei Gu, Jing Liu, Linjie Luo,
arXiv:2509.15496
   * Lynx: Enabling Efficient MoE Inference through Dynamic Batch-Aware Expert Selection,
Vima Gupta, Jae Hyung Ju, Kartik Sinha, Ada Gavrilovska, Anand Padmanabha Iyer,
arXiv:2411.08982
   * Lynx: An Open Source Hallucination Evaluation Model,
Selvan Sunitha Ravi, Bartosz Mielczarek, Anand Kannappan, Douwe Kiela, Rebecca Qian,
arXiv:2407.08488
   * LYNX: Learning Dynamic Exits for Confidence-Controlled Reasoning,
Ömer Faruk Akgül, Yusuf Hakan Kalaycı, Rajgopal Kannan, Willie Neiswanger, Viktor Prasanna,
arXiv:2512.05325
   * What Matters in Training a GPT4-Style Language Model with Multimodal Inputs?
Yan Zeng, Hanbo Zhang, Jiani Zheng, Jiangnan Xia, Guoqiang Wei, Yang Wei, Yuchen Zhang, Tao Kong,
arXiv:2307.02469

---

> ### Author Rebuttal · Authors · 2026-03-31
>
> **[Q1] Novelty and contribution**
>
> We clarify that our contribution is not unpaired alignment alone. The paper contributes four points:
>
> (1) empirical evidence that pretrained Video LLMs contain a reusable video-induced token interface;
>
> (2) a shared-path modality adaptation framework that reuses the frozen visual pathway instead of adding a heavy target-modality encoder;
>
> (3) an interface alignment objective combining attention-response alignment and statistical regularization using only unpaired unimodal data; and
>
> (4) extensive validation across audio, 3D, high-frame-rate, and egocentric video, showing stronger accuracy-efficiency trade-offs than heavier patching-based methods such as PAVE.
>
> We will revise the paper to present this contribution more clearly.
>
> **[Q2] Relationship to the Platonic representation hypothesis**
>
> We thank the reviewer for this insightful point. Our work is related in spirit to the Platonic representation hypothesis, since both suggest shared structural regularities across modalities, which is consistent with our token-interface view. However, LynX makes a much narrower and more operational claim. We do not assume a universal modality-agnostic latent space across models or modalities. Instead, we focus on a model-specific, video-induced token interface already internalized by a pretrained Video LLM. Accordingly, LynX aligns new modalities not to an abstract universal space, but to the concrete operating regime of the frozen visual pathway, namely its attention behavior and projector-level token statistics. In this sense, the Platonic hypothesis motivates why such an interface may exist, while LynX provides a practical mechanism to exploit it without disrupting the model’s original video-language capability. We will clarify this connection in the revised manuscript.
>
> **[Q3] Relationship to 'Zhang et al. (2023)'**
>
> We emphasize that LynX and Zhang et al. (2023) address different problems. Zhang et al. study robustness to unseen modality combinations at inference time, using a common latent space, summation-based aggregation, and pseudo-supervision for modality reliability. In contrast, LynX addresses pretrained Video LLM expansion: integrating a new modality without adding a heavy modality-specific encoder or complex fusion stack, and without paired target-modality/video/text supervision. The two methods therefore differ in objective (unseen-combination robustness vs. new-modality integration), training signal (pseudo-supervision vs. interface compatibility), and mechanism (common-space projection vs. reuse of the existing video pathway).
>
> **[Q4] Role of `interface guidance'**
>
> We believe the confusion comes from the role of interface guidance, so we clarify it below step by step. We would be happy to provide further clarification if needed.
>
> (1) Interface guidance.
>
> Interface guidance defines the native video operating regime of the pre-trained Video LLM. From unlabeled reference videos $\mathcal{V}$, LynX computes: (i) layer-wise Key/Value references $\{K_v^{(l)}, V_v^{(l)}\}$ in the vision encoder (Eq. (2)), and (ii) the mean and variance ($\mu_v, \sigma_v^2$) of projected video tokens.
>
> (2) How does attention-response alignment achieve interface alignment?
>
> During alignment, the new modality is passed through the same frozen vision encoder with LoRAs, and the LoRAs are optimized so that the new modality follows the same internal processing pattern as native video. In this sense, interface guidance provides a stable anchor, and attention-response alignment maps the new modality to that anchor.
>
> (3) Why are videos used before alignment, and why is this separate from evaluation?
>
> The reference videos are used only to estimate these video-native statistics. They are unlabeled, unpaired, and independent of benchmark supervision, so this step is separate from evaluation or task training. It is a reference-estimation step, not an additional supervised stage.
>
> (4) Interpretation of Table 7.
>
> Under this interpretation, the small differences in Table 7 are desirable. They indicate that the estimated interface is stable across different reference subsets, rather than brittle or dataset-specific. Thus, Table 7 supports the role of interface guidance as a robust anchor for the pre-trained video pathway.
>
> If any part of the method remains unclear, we would be happy to further clarify and provide additional details.
>
> **[Minor] Name of 'LynX'**
>
> We thank the reviewer for the suggestion. We agree that the name may not be sufficiently distinctive, and we will consider revising it to avoid potential confusion with prior works.
>
> **[L1] Limitation**
>
> We provided the limitation regarding the inherent and modality-specific transformation in Appendix C; please refer to it for details.

---

> > ### Author Rebuttal · Reviewer_HBTt · 2026-04-02
> >
> > Authors have partially resolved my concerns and I will update my score. What is still lacking for me is insight, what constitutes the main driver for the improvement, and why? Would be valuable for the paper if the authors could strengthen this aspect.

---

> > > ### Author Response · Authors · 2026-04-04
> > >
> > > We sincerely thank the reviewer for the willingness to increase the score and for this valuable suggestion.
> > >
> > > Our current evidence supports the following mechanism: the dominant source of improvement is **reusing pre-trained video encoder parameters** to build a novel pathway for a new modality, with attention response alignment as the primary component and distribution regularization playing a secondary but important stabilizing role.
> > >
> > > Specifically, the difficulty in *adding a new modality* is an off-manifold problem: the new modality initially induces a kernel and encoder dynamics that are misaligned with those of the native modality.
> > > In contrast to existing works for employing a new modality encoder and a new projector for LLM, our idea is to mimic the video-induced kernel structure for the new modality process.
> > > By doing so, we address this problem as an on-manifold one.
> > > As the reviewer suggested, this view is consistent with the Platonic representation hypothesis.
> > > From this viewpoint, alignment should be achieved at the level of induced similarity or kernel structure (i.e., attention-response alignment in our method), rather than through exact point-wise coordinate matching.
> > >
> > > To sum up, LynX aligns a new modality to the video-induced relational geometry of the pre-trained model, while keeping the resulting token distribution compatible with downstream processing.
> > >
> > > We will include this discussion in the revised paper, and also welcome the reviewer to follow up in further discussion.
> > > Thank you again for the valuable feedback that has helped strengthen our paper.
> > >
> > > **We sincerely thank the reviewer for the careful reading and valuable discussion.**
> > >
> > > During the rebuttal period, we worked to directly address the key concerns through additional experiments, stronger analysis, and clearer explanation of the method and its main driver. We believe these additions substantially strengthen the paper, especially in clarifying the novelty, the mechanism behind the improvement, and the generality of the approach across backbones and modalities. We hope the reviewer finds the revised evidence convincing, and if so, we would greatly appreciate reconsideration of the current rating.

---

### Official Review · Reviewer_LnaB · 2026-03-09

**Soundness:** 3
**Presentation:** 3
**Significance:** 3
**Originality:** 3
**Overall Recommendation:** 5
**Confidence:** 3

**Summary:**

LynX exploits an internal token-manifold interface in Video LLMs to add new modalities without heavy encoders or paired supervision. It attaches a lightweight auxiliary branch to the frozen vision encoder and aligns attention patterns and feature distributions using unimodal data, preserving intrinsic video priors. Experiments show state-of-the-art accuracy and efficiency across audio-visual QA, 3D reasoning, high-frame-rate, and multi-view video tasks.

**Compliance With Llm Reviewing Policy:**

Affirmed.

**Final Justification:**

The rebuttal addresses most of my questions, particularly the clarification on manifold compatibility, and I am inclined to raise my score.

**Key Questions For Authors:**

Please answer the questions raised above.

**Limitations:**

yes

**Strengths And Weaknesses:**

The manuscript is well organized and easy to follow. LynX extends Video LLMs to new modalities via a lightweight unimodal alignment branch, achieving state-of-the-art accuracy and efficiency across diverse video understanding tasks. However, I have several questions regarding this manuscript:
1. The proposed method shows varying levels of improvement across benchmarks, as shown in Fig. 1 (a). It would be helpful to discuss further these differences in light of each dataset’s characteristics and explain why the gains are larger in some cases than others.
2. The manuscript claims that distributional alignment ensures manifold compatibility while preserving the integrity of the Video LLM. However, the current analysis is not sufficiently insightful. Please provide a more detailed and convincing discussion of what “manifold compatibility” means in this context and how distributional alignment achieves it.
3. The experiments mainly use LLaVA and Qwen2. It would be helpful to evaluate the method with additional LLM backbones to better demonstrate its generalization ability.
4.Could the authors provide visualizations (e.g., attention maps) before and after aligning attention responses across different modalities to better illustrate the effect of the proposed alignment?

---

> ### Author Rebuttal · Authors · 2026-03-31
>
> **[Q1] Different impact on each task**
>
> We thank the reviewer for this helpful question. Our LynX is most effective when the target modality modifies how the scene
> should be represented while remaining compatible with the video-induced interface. This is why the gains are larger on 3D
> QA, high-frame-rate video, and multi-view video reasoning: these tasks change geometry, temporal density, or viewpoint,
> but the resulting signal still maps naturally into the representational regime already learned by the video pathway. Gains
> depend on whether RGB is already sufficient or when the target signal is only weakly expressible through that interface.
> The improvements are also more pronounced for the smaller model (i.e., LynX-0.5B) than for the larger model (i.e.,
> LynX-7B). We attribute this to two factors: (1) the smaller backbone has a larger initial modality gap and therefore benefits
> more from explicitly aligning the new modality to the pretrained video interface, and (2) the relative adaptation capacity
> is effectively larger, i.e., the injected trainable parameters (i.e., LoRAs) constitute a more influential correction for a
> smaller backbone than for a much larger one. In contrast, the larger model already possesses stronger native reasoning
> and representation capacity, so the same interface-alignment mechanism still improves performance, but the relative gain
> appears smaller because the baseline is already stronger. Overall, this trend is consistent with the role of LynX: it is most
> beneficial when the pretrained model needs stronger help to translate a new modality into its native video token regime.
>
>
> **[Q2] Manifold compatibility**
>
> In our paper, manifold compatibility does not mean that new-modality tokens should overlap with the ordinary vocabulary
> manifold. It means that they should be mapped into the native token regime already induced by the pre-trained visual
> pathway, so that the frozen projector and LLM can process them without changing their original operating behavior. In other
> words, compatibility is with the pre-existing visual token interface of the Video LLM.
>
> We further make this point more specifically through quantitative analysis of the frozen embedding space.
> Specifically, we quantitatively analyze the embedding space using three statistics: (1) the averaged pairwise cosine distance, (2) the L2 norm of the modality-wise mean embedding, and (3) a scale-invariant modality gap (Liang et al., 2022) between frame and vocabulary embeddings.
> We recommend referring to our response on zrZE's [W1 & Q1].
>
> **[Q3] Broader baseline comparison**
>
> We thank the reviewer for pointing this out.
> For a direct and fair comparison with PAVE under the same backbone family, we used LLaVA-OV-0.5B and -7B.
> Importantly, our method is not limited to LLaVA.
> To further demonstrate the effectiveness and generalizability, we additionally evaluate LynX with a different Video LLM, Qwen2.5-VL-3B, on SQA3D.
>
> | Method | SQA3D EM@1 | ΔParams. |
> | --- | --- | --- |
> | **Zero-shot Video LLMs** |  |  |
> | LLaVA-OV-0.5B | 0.8 (43.0) | - |
> | Qwen2.5-VL-3B | 15.1 | - |
> | **Task-specific models < 7B** |  |  |
> | LynX-LLaVA-OV-0.5B (**Ours**) | 52.2 (54.2) | 68.7M |
> | **LynX-Qwen2.5-VL-3B (Ours)** | **59.7 (62.0)** | 165.5M |
>
> As shown in the above table, LynX with Qwen2.5-VL-3B achieves 59.7EM@1, substantially improving over the zero-shot Qwen2.5-VL-3B baseline by 44.6 points.
> Importantly, this result also exceeds the performance of LynX-LLaVA-OV-0.5B, showing that the main takeaway of the paper does not rely only on comparisons within the LLaVA family.
>
> Due to the limited rebuttal period, we report preliminary results with Qwen2.5-VL-3B on SQA3D.
> In the revised manuscript, we will extend this analysis to additional backbones, including Qwen2.5-VL-7B, InterVL2.5 (4B, 8B), and provide comprehensive evaluations across all benchmarks.
>
> **[Q4] Attention**
>
> We compare the attention visualization before and after interface alignment. Specifically, we extract attention maps
> between question tokens and the 3D tokens (from the pre-trained vision encoder without LoRAs) for the same example in
> Figure 4 of the main paper. The results show that the model before interface alignment attends to random and question-
> irrelevant regions in the depth maps, failing to focus on the target object for reasoning. We will include the visualization
> comparison in the revised version.

---

> > ### Author Rebuttal · Reviewer_LnaB · 2026-04-04
> >
> > The rebuttal addresses most of my questions, especially the explanations about the manifold compatibility. I am inclined to raise my score.

---

> > > ### Author Response · Authors · 2026-04-04
> > >
> > > We sincerely thank the reviewer for the continued support of our work. Your detailed and constructive feedback throughout the review process has meaningfully improved the paper.

---

### Official Review · Reviewer_LjZu · 2026-03-13

**Soundness:** 3
**Presentation:** 3
**Significance:** 3
**Originality:** 3
**Overall Recommendation:** 4
**Confidence:** 4

**Summary:**

The paper studies how Video LLMs process visual tokens internally and observes that the vision encoder + projector pipeline produces embeddings that occupy a continuous geometric region distinct from the vocabulary embedding space. The authors call this region the “token interface.” Building on this, they propose LynX, a method for extending Video LLMs to new modalities by inserting LoRA adapters into the frozen vision encoder. After alignment, standard instruction tuning with LoRA in the LLM is applied. Experiments on audio-visual QA, 3D QA, video understanding, and multi-view proficiency estimation show consistent improvements over LLaVA-OV baselines and PAVE, with substantially fewer additional parameters.

**Compliance With Llm Reviewing Policy:**

Affirmed.

**Final Justification:**

Thanks for the authors' rebuttal. My concerns have been adequately addressed.

**Key Questions For Authors:**

Q1. Have you tried replacing the mean/variance matching with a richer distributional alignment, such as MMD or sliced-Wasserstein distance? The near-zero ablation effect of L_stat in Table 5 suggests the current formulation may be doing very little. If a stronger distributional constraint improves results, that would support the manifold-alignment narrative.
Q2. What is the variance of the averaged K/V references across the reference set? If variance is high, the mean is a poor representative, and per-cluster or per-category references might work better. If the variance is low and the mean is stable, the current approach is well-justified.
Q3. For the MUSIC-AVQA failure cases (Figure 6c), both LynX and the baseline fail on purely audio-driven questions. Have you measured what fraction of the test set falls into this category, and what LynX’s accuracy is on the audio-only subset specifically? This would quantify the severity of the inherent limitation discussed in Appendix C.

**Limitations:**

Yes

**Strengths And Weaknesses:**

The paper evaluates across four distinct modality types and nine benchmarks. The consistent gains across all of them lend credibility to the claim that the approach is general.

---

> ### Author Rebuttal · Authors · 2026-03-31
>
> **[Q1] Distributional metrics (e.g., MMD, sliced-Wasserstein)**
>
> We appreciate your valuable comment. We analyze the performance with a richer projector-level distribution regularizer, including MMD and sliced Wasserstein distance, in place of the current MSE objective.
>
> | Regularization | SQA3D EM@1 |
> | --- | --- |
> | MSE | 52.2 |
> | MMD | 52.4 |
> | s-Wasserstein | 52.3 |
>
> The results show that these alternatives provide only marginal differences from the current formulation on SQA3D: MSE achieves 52.2 EM@1, while MMD and sliced-Wasserstein yield 52.4 and 52.3, respectively.
> This result is consistent with the role of $\mathcal{L}\text{stat}$ in LynX, suggesting that $\mathcal{L}\text{stat}$ is not to serve as the primary adaptation mechanism, but to regularize the projector-level token distribution.
>
> In addition, we empirically observe that these richer objectives are more sensitive to hyperparameters and batch statistics, while the current MSE regularization is simple, stable, and computationally light. We will include this analysis in the manuscript.
>
> **[Q2] Stability of averaged K/V references**
>
> We agree that this is an important point and we will clarify it.
> While we did not explicitly report variance statistics, Table 7 already indicates that performance is highly consistent across different reference sources (audio, 3D, video, and mixed).
> We further measure the variance of the averaged Key/Value references across the reference videos.
>
> | Component | Avg. |
> | --- | --- |
> | Key Mean | -0.0133425555 |
> | Key Variance | 0.0020495121 |
> | Value Mean | 0.0011632288 |
> | Value Variance | 0.0001416960 |
>
> The result indicates that the global reference is stable: when averaged across layers, the variance is 0.00205 for the Key and 0.00014 for the Value, while the corresponding means are -0.01334 and 0.00116, respectively.
> These small variances indicate that the reference K/V statistics are tightly concentrated across videos rather than being dominated by large sample-to-sample fluctuations.
>
> Due to the rebuttal space limit, we report the values averaged over layers here.
> If helpful, we are happy to provide layer-wise results and detailed breakdowns to further support this analysis.
>
> **[Q3] Failure case on MUSIC-AVQA**
>
> In MUSIC-AVQA, the number of audio-biased questions is 1,623 out of 9,185, which corresponds to 17.7\% of the test set.
> For this subset, LynX-7B achieves 80.8\% accuracy, as shown in Table 1; the corresponding score for LynX-0.5B is 78.9\%.
> They correspond to a hard subset in which the target source is not visually grounded at all and must be inferred from subtle background audio cues alone, as described in Appendix C.
> Otherwise, on the visual subsets, with 93.5\% for LynX-7B and 92.2\% for LynX-0.5B, respectively.
> Overall, the quantitative result shows that LynX remains effective on the audio-only portion of MUSIC-AVQA, while the limitation is concentrated in particularly challenging cases where the answer depends almost entirely on non-visual evidence.

---

> > ### Author Rebuttal · Reviewer_LjZu · 2026-04-04
> >
> > My concerns have been adequately addressed.

---

> > > ### Author Response · Authors · 2026-04-05
> > >
> > > We sincerely thank the reviewer for reconsidering the score. Your detailed and constructive feedback throughout the review process will contribute meaningfully to improving the paper.

---

### Official Review · Reviewer_zrZe · 2026-03-13

**Soundness:** 3
**Presentation:** 3
**Significance:** 3
**Originality:** 4
**Overall Recommendation:** 4
**Confidence:** 3

**Summary:**

The paper studies how to extend a pretrained Video LLM to new modalities such as audio, 3D, high-frame-rate video, and egocentric video without training heavy modality-specific encoders or requiring paired multimodal supervision.
The authors argue that the pretrained video pathway induces a continuous manifold in the LLM token space, and that new modalities can be mapped into this interface rather than aligned through a separate multimodal encoder stack.
The proposed method, LynX, adds LoRA adapters to the frozen vision encoder for new modalities and trains them using only unimodal data with:
- attention response alignment to match video-derived attention behavior,
- distribution regularization to match mean/variance of projected video tokens,

followed by LoRA-based instruction tuning on the LLM.

**Compliance With Llm Reviewing Policy:**

Affirmed.

**Final Justification:**

I retain my support for the work. It has an interesting lens toward the problem.

**Key Questions For Authors:**

- Can you provide stronger evidence for the "token interface" claim beyond t-SNE and downstream performance?
- How well would LynX handle a modality that is much less visually grounded than the ones tested here?

**Limitations:**

Yes, the authors discuss meaningful limitations of their work (e.g. failure on concepts not grounded in vision).

**Strengths And Weaknesses:**

### Strengths
- Interesting and original perspective:
   - The "token interface" view is a compelling framing and new to me.
   - The t-SNE visualization in Figure 2 supports the claim that frame embeddings occupy a distinct region from vocabulary embeddings, motivating a reusable latent interface.
- Method is simple and practical:
   - The approach avoids training a full modality-specific encoder + fusion stack.
   - It only adds LoRAs in the vision encoder and then LoRAs in the LLM, which is architecturally clean.
- Good performance with much lower added parameters: Since LynX escape the process of building separated encoder, the overhead is kept mininum (Table 1,2,3,4).

### Weaknesses
- Even though "token interface" claim is interesting, it is not established: The evidence is suggestive but indirect--via t-SNE, downstream task performance and attention-map. There is not yet a rigorous demonstration that a stable, reusable manifold truly exists as a model inherent property.
- Comparisons are centered mostly on LLaVA family: Both PAVE and LLaVA-OV-FT are backed by a same model family (LLaVA), so the result could be biased. For some tasks, comparisons to broader modality-specific methods exist, but the main takeaways still rely heavily on beating PAVE.

---

> ### Author Rebuttal · Authors · 2026-03-31
>
> **[W1 & Q1] Evidence for the “token interface”**
>
> We thank the reviewer for the thoughtful feedback. We further quantitatively analyze the embedding space using three statistics: (1) the averaged pairwise cosine distance, (2) the L2 norm of the modality-wise mean embedding, and (3) a scale-invariant modality gap (Liang et al., 2022) between frame and
> vocabulary embeddings.
>
> | Model | Vocab.-Vocab. | Frame-Frame | Vocab.-Frame | Vocab. (L2) | Frame (L2) | Δ_gap    |
> |--------------------|--------------|-------------|--------------|-------------|------------|----------|
> | LLaVA-OV-0.5B | 0.71 | 0.10 | 0.96 | 0.19 | 26.30 | 1.0081   |
> | LLaVA-OV-7B | 0.78  | 0.11 | 0.99 | 0.10 | 45.42 | 0.9499   |
> | Qwen2.5-VL-3B | 0.79  | 0.35 | 1.02 | 0.47 | 31.65 | 0.9539   |
> | InterVL2.5-4B | 0.92 | 0.11 | 1.01 | 0.29 | 41.73 | 0.9930   |
>
> The results consistently show that:
> -  frame embeddings from a significantly more compact geometric regime than vocabulary embeddings,
>
> - the cosine distance between frame and vocabulary embeddings is close to orthogonal, indicating clear separation rather than intermingling, and
>
> - the modality gap remains large even after normalization, ruling out trivial scale effects.
>
> The key point is that this region is not merely separate, but functionally consumable by the frozen projector and LLM stack.
> This is supported by our empirical findings that (i) new modalities can be mapped into this region without paired cross-modal supervision in the alignment stage (Table 1-4 and Figure 4), and (ii) removing interface alignment leads to a substantial performance drop (Table 5).
>
> Furthermore, we observe the same geometric patterns across multiple architectures beyond LLaVA series, including Qwen2.5-VL and InternVL2.5, suggesting that this phenomenon is not model-specific but a general property of pretrained Vision LLMs. We will include these additional analyses and results in the revised manuscript.
>
> **[W2] Broader baseline comparison**
>
> We thank the reviewer for pointing this out.
> For a direct and fair comparison with PAVE under the same backbone family, we used LLaVA series.
> Importantly, our method is not limited to LLaVA.
> To further demonstrate the effectiveness and generalizability, we additionally evaluate LynX with a different Video LLM, Qwen2.5-VL-3B, on SQA3D.
>
> | Method | SQA3D EM@1 | Δ Params |
> |---------------------------------|--------------------|----------|
> | **Zero-shot Video LLMs** |  | |
> | LLaVA-OV-0.5B  | 0.8 (43.0) | - |
> | Qwen2.5-VL-3B  | 15.1 | -  |
> | **Task-specific models < 7B**  | | |
> | LynX-LLaVA-OV-0.5B (Ours) | 52.2 (54.2) | 68.7M |
> | LynX-Qwen2.5-VL-3B (Ours) | **59.7 (62.0)** | 165.5M |
>
> LynX with Qwen2.5-VL-3B achieves 59.7EM@1, substantially improving over the zero-shot Qwen2.5-VL-3B baseline by 44.6 points.
> Importantly, this result also exceeds the performance of LynX-LLaVA-OV-0.5B (52.2EM@1), showing that the main takeaway of the paper does not rely only on comparisons within the LLaVA family.
>
> Due to the limited rebuttal period, we report preliminary results with Qwen2.5-VL-3B on SQA3D.
> In the revised manuscript, we will extend this analysis to additional backbones, including Qwen2.5-VL-7B, InterVL2.5 (4B, 8B), and provide comprehensive evaluations across all benchmarks.
>
> **[Q2] Less visually grounded modality**
>
> To directly verify the effectiveness of our LynX on less visually grounded task, we conduct an additional experiment on AVUT (Yang et al., 2025), an audio-centric benchmark that is substantially less visually grounded.
> Specifically, we train the model on AV-Gemini and evaluate on AV-Human.
>
> | Method                  | AV-Human Acc. (%) |
> |-------------------------|-------------------|
> | **Visual MLLMs**        |  |
> | Qwen2-VL-7B            | 58.38   |
> | LLaVA-Video-7B         | 56.52 |
> | InternVL2-8B           | 45.9   |
> | VideoLLaVA-7B          | 33.14  |
> | **Audio MLLMs**        |    |
> | SALMONN-13B            | 36.48  |
> | **Audio-visual MLLMs** |  |
> | VideoLLaMA2-7B         | 44.90 |
> | video-SALMONN-13B  | 38.33 |
> | PandaGPT-13B   | 25.38 |
> | LynX-LLaVA-OV-0.5B  | 46.91 |
>
> As shown in the table, LynX achieves comparable performance to several larger baselines and even better performance than InternVL2-8B, VideoLLaVA-7B, SALMONN-13B, VideoLLaMA2-7B, video-SALMONN-13B, and PandaGPT-13B. This shows that LynX does not collapse when the modality becomes substantially less visually grounded; the video-induced interface still transfers non-trivially to audio-centric reasoning.
>
> At the same time, the gap to the strongest audio-visual frontier model on this benchmark indicates a clear boundary condition: LynX is strongest when the new modality remains at least partially expressible through the pretrained visual interface, and weaker when the task depends predominantly on modality-specific evidence with little visual correlate. This additional AVUT analysis will be included in the revised manuscript, together with the quantified MUSIC-AVQA audio-only analysis.

---

> > ### Author Rebuttal · Reviewer_zrZe · 2026-04-02
> >
> > Thanks the authors for the detailed response. Most of my concerns are clear, I would retain my score.

---

> > > ### Author Response · Authors · 2026-04-04
> > >
> > > We sincerely thank the reviewer for the continued support of our work. Your detailed and constructive feedback throughout the review process has meaningfully improved the paper.

---

### Decision · Program_Chairs · 2026-04-30

**Decision:**

Accept (regular)

**Comment:**

## Summary
This paper proposes Lynx, a scalable framework that integrates novel modalities by repurposing the token-representation of video-LLMs.

## Ratings
Initially, the paper receives 3 accept recommendations and 1 weak reject recommendation. After rebuttal, all reviewers are convinced and thus all recommend accepting the paper.

## Discussion and Decision
AC reads all reviews and discussions. AC agrees with the reviewers thus recommends accepting the paper.